

**Circulation of the European Northwest Shelf: A Lagrangian perspective**
**Marcel Ricker[1]* and Emil V. Stanev[2]**
[1]University of Oldenburg, Institute for Chemistry and Biology of the Marine Environment, Carl-von-
Ossietzky-Straße 9-11, 26111 Oldenburg, Germany
[2]Institute of Coastal Research, Helmholtz-Zentrum Geesthacht, Max-Planck-Straße 1, 21502
Geesthacht, Germany
*Corresponding author: E-mail: marcel.ricker@uni-oldenburg.de



**Abstract**
The dynamics of the European Northwest Shelf (ENWS), the surrounding deep ocean, and the
continental slope between them are analysed in a framework of numerical simulations using Lagrangian
methods. Several sensitivity experiments are carried out in which (1) the tides are switched off, (2) the
wind forcing is low-pass filtered, and (3) the wind forcing is switched off. To measure particle
accumulation, a quantity named the "density trend" is introduced. Yearly averages of the results in the
deep ocean show no permanent particle accumulation areas at the surface. On the shelf, elongated
accumulation patterns persist on yearly time scales, often occurring along the thermohaline fronts. In
contrast, monthly accumulation patterns are highly variable in both regimes. Tides substantially affect
the particle dynamics on the shelf and thus the positions of fronts. The contribution of wind to particle
accumulation in specific regions is comparable to that of tides. The role of vertical movements in the
dynamics of Lagrangian particles is quantified both for the eddy-dominated deep ocean and for the
shallow shelf. In the latter area, winds normal to coasts result in upwelling and downwelling and very
clear patterns characterising the accumulation of Lagrangian particles associated with the vertical
circulations.



## 1 Introduction

The European Northwest Shelf (ENWS) (Fig. 1) is among the most studied ocean areas worldwide.
Numerous reviews have presented details of its physical oceanography (e.g. Otto et al., 1990;
Huthnance, 1991 Understanding the dynamics of the ENWS has been achieved considerably through
numerical modeling (Maier-Reimer, 1977; Backhaus, 1979; Heaps, 1980; Davies et al., 1985; Holt and
James, 1999; Pohlmann, 2006; Zhang et al., 2016; Pätsch et al., 2017). In contrast, the usefulness of
Lagrangian methods for better understanding the ENWS dynamics has not been widely investigated
heretofore.
In the following, we briefly summarize some basic oceanographic knowledge about the ENWS (the
study area is shown in Fig. 1). The slope current dynamics and exchanges between the deep ocean and
shelf have been analysed by Huthnance (1995), Davies and Xing (2001), Huthnance et al. (2009) and
Marsh et al. (2017); Lagrangian drifter experiments in this area have been described by, e.g. Booth
(1988) and Porter et al. (2016). The prevailing westerlies induce on-shelf water transport from the Celtic
Sea up to the Outer Hebrides (Huthnance et al., 2009). Water entering the Celtic Sea flows either into
the English Channel or into the Irish Sea via St. George's Channel. The water exiting the Irish Sea flows
around the Outer Hebrides and joins the on-shelf transported water. Part of this water enters the North
Sea, mainly via the Fair Isle Current, where it begins an anti-clockwise journey through the North Sea.
The third path of waters entering the North Sea originates from the Baltic, following which those waters
are integrated into a complex system of currents in the Skagerrak and the Norwegian Trench. In this
area, the Atlantic and Baltic Sea waters undergo strong mixing. Along the southern slope of the
Norwegian Trench, a branch of the European Slope Current flows toward the Baltic Sea, while a current
flowing in the opposite direction follows the northern slope of the trench. In addition, large river runoff
influences the water masses in the North Sea and along the Scandinavian coast, explaining the low
salinity along coastal areas. The major hypothesis in the present study is that although the North Sea is
very shallow, it contains an important vertical circulation. Revealing such characteristics is the first
objective of the present study.



Much is known about the thermohaline fronts on the ENWS and its estuaries (Simpson and Hunter,
1974; Hill et al., 2008; Holt and Umlauf, 2008; Pietrzak et al., 2011). Although large parts of this ocean
area are vertically well mixed, seasonal and shorter-term variability lead to pronounced differences in
the positions and strengths of the fronts; freshwater fluxes are also important, particularly in shallow
coastal areas. Krause et al. (1986), Le Fèvre (1986), Belkin et al. (2009), Lohmann and Belkin (2014),
Mahadevan (2016) and McWilliams (2016) addressed the biological consequences of frontal systems,
and the frontal physics are summarized in Simpson and Sharples (2012). However, to the best of the
authors' knowledge, the frontal dynamics of the ENWS have not been addressed from a Lagrangian
perspective, which constitutes the second objective of our study.

Most previous studies that employed Lagrangian particle tracking in the region of the ENWS
(Backhaus, 1985; Hainbucher et al., 1987; Schönfeld, 1995; Rolinski, 1999; Daewel et al., 2008; Callies
et al., 2011; Neumann et al., 2014 and Marsh et al., 2017) addressed only part of the region studied
herein. Hence, our third objective is to provide a comparison among the specific hydrodynamic regimes
in different areas of the ENWS and exchanges between these areas. One example has been recently
provided by Marsh et al. (2017) for part of the European Slope Current.

The present study was initiated in the framework of a project studying the fate of marine litter in
the North Sea (Gutow et al, 2018; Stanev et al, 2019). Here, we extend the area of our analyses to
include the entire ENWS, the European Slope Current, the Bay of Biscay and parts of the Northeast
Atlantic. Unlike our recent studies, herein, we address Lagrangian particles transported by 3-D ocean
currents (Stokes drift and wind drag are not considered). Lagrangian approaches applied to the study of
other ocean regions can be found in, e.g. Bower et al. (2009) for the North Atlantic, Paparella et al.
(1997) for the Antarctic Circumpolar Current, Reisser et al. (2013) for Australia, van Sebille et al.
(2015) for the world ocean, Maximenko et al. (2018) for tsunamis, Froyland et al. (2014) and van der
Molen et al. (2018) in terms of connectivity studies.



In Sect. 2, we will describe the model, its setup, the simulated dynamics and the Lagrangian
experiments. In Sect. 3, our results are presented and discussed. The paper ends with a brief conclusion
and an outlook in Sect. 4.

**2 Materials and methods**
*2.1 The numerical model*
The Nucleus for European Modelling of the Ocean (NEMO) hydrodynamic ocean model is used in this
paper (https://www.nemo-ocean.eu/; Madec, 2008). For this study, the Atlantic Margin Model
configuration with a 7 km resolution (AMM7; Fig. 1) of NEMO is chosen because it appears to be one
of the best validated models for the ENWS (O'Dea et al., 2012 and 2017). The numerical model solves
the primitive equations using hydrostatic and Boussinesq approximations. The horizontal resolution is
$^1/_9°$ in the zonal direction and $^1/_{15}°$ in the meridional direction; that is, the resolution is approximately
7.4 km. There are $297 \times 375$ grid points altogether and 51 vertical $\sigma$-layers. For tracer advection, we
employ the total variation diminishing (TVD) scheme, diffusion takes place on geopotential levels with
a Laplacian operator (the horizontal eddy diffusivity is specified as 50 $m^2$ $s^{-1}$). For momentum diffusion,
a bi-Laplacian scheme is applied to act on the model levels (horizontal eddy viscosity = $-1·10^{10}$ $m^4$ $s^{-1}$).
The generic length scale (GLS) $k$-$\varepsilon$ scheme is used as the turbulence closure scheme; the bottom friction
is nonlinear with a log-layer structure and a minimum drag coefficient of $1·10^{-3}$. The baroclinic time
step is 300 s. The output, including the salinity (S), temperature (T), velocities (u, v) and sea surface
height (SSH), is written hourly.

The atmospheric forcing is provided by the UK Met Office atmospheric model with a 3-h temporal
resolution for the fluxes and an hourly resolution for the 10-m wind and air pressure. The model uses
climatological river runoff, and tidal forcing is prescribed at the open boundary. The period of
integration considered here spans from 01 January 2014 to 31 December 2015 of which the first year is
the spin-up period. The analyses of the results are performed for the area between 42.57° N–63.50° N
and 17.59° W–13.00° E.



Although part of this study could be performed using the freely available Forecasting Ocean
Assimilation Model (FOAM) AMM7 data (https://marine.copernicus.eu), we run the abovementioned
model to (1) perform Lagrangian simulations online and (2) carry out some additional sensitivity
experiments. In contrast to the operational AMM7 model, the data are not assimilated here. In the
following, the basic experiment is referred to as the control run (CR). In one sensitivity experiment, the
tides are turned off; this experiment is referred to hereafter as the nontidal experiment (NTE). In two
other sensitivity experiments, the wind forcing is low-pass filtered with a moving time window of one
week (referred to as the filtered-wind experiment, FWE) or completely turned off (the nonwind
experiment, NWE).

*2.2. Model velocity validation*
The velocities of the CR have been validated using 9 passive GPS surface drifters; these drifters provide
the most appropriate type of in situ data for validating the model's ability of particle advection. The
drifters, which have a bottom-mounted sail to reduce direct wind drag, were designed to be moved by
the upper 1 m of the ocean. The drifters were released in the German Bight during RV *Heincke* cruise
HE445, and their position was sent every ~20 minutes from May to July 2015. The dataset is freely
available (Carrasco and Horstmann, 2015). To the best of the authors' knowledge, this is the only GPS
drifter dataset available for the ENWS during the period of the simulations analysed herein. For
validation, the model velocities are interpolated to the drifter positions in space and time and compared
against the drifter velocities. The corresponding scatter plots (Fig. S1a and S1b) show a good model
performance in the range of ±25 cm s$^{-1}$, where the quantile-quantile plot (qq-plot) is almost along the
diagonal. Deficiencies in the model occur at higher velocities, where the model is too slow.
Nevertheless, the linear correlations of the u and v velocity components of 0.89 and 0.85, respectively,
between the drifters and the model and the corresponding root mean square errors (RMSEs) of 13.9 and
12.2 cm s$^{-1}$ are considered to reflect a satisfactory model performance (Table 1).

The quality of the above numbers illustrating the model skill can be better understood if the drifter
data are compared with independent observations; in the following, a comparison is performed with HF



radar data. The HF radar system described by Stanev et al. (2015) and Baschek et al. (2017) consists of
3 measurement stations covering most of the German Bight and measures ocean surface velocities.
These data are freely available (http://codm.hzg.de/codm/). The corresponding scatter plots (Fig. S1c
and S1d) do not show as much underestimation of high velocities as in the model (compare with Fig.
S1a and S1b), but the spread of the data in two observations is comparable to the case of the model-
data comparison (the standard deviation between the two observations is even larger than in the case of
the model-data comparison). The conclusion from Table 1 is that the difference between the estimations
from the model and data are not larger than that between two observations. Similar validations provided
by Stanev et al. (2019) for the North Sea also demonstrate the credibility of the Lagrangian tracking
approach.

**Table 1**: Summary of the model validation performed by comparing GPS drifter velocities with the CR
and HF radar velocities; the surface velocity components of the latter were interpolated to the drifter
velocities. Details are given in the text. A positive bias denotes that drifter velocities are larger than the
velocities of the CR or HF radar. The corresponding scatter plots are given in Fig. S1.

|  | Drifter - CR | | Drifter - HF radar | |
|---|---|---|---|---|
|  | u | v | u | V |
| RMSE [cm s⁻¹] | 13.9 | 12.2 | 18.4 | 12.5 |
| Linear correlation | 0.89 | 0.85 | 0.91 | 0.87 |
| Standard deviation [cm s⁻¹] | 13.7 | 12.1 | 16.9 | 12.4 |
| Bias [cm s⁻¹] | 2.7 | 1.3 | 7.4 | -2.0 |


*2.3. Analysis of the simulated dynamics*
The circulation of the CR is very diverse in different model areas, but the differences among the
dynamic regimes in the CR are most pronounced between the deep ocean and the shelf. The averaged
velocities and velocity amplitudes (U) for January 2015 (Fig. 2a and 2b) show basically two regimes:
an eddy-dominated regime west of the continental slope and a tidally dominated regime on the shelf.
The latter is characterized by relatively low mean velocities (Fig. 2a) and large velocity oscillations in
the English Channel (Fig. 2b). The transition between these two regimes occurs along the 200 m isobath
(Fig. 2a), which can be considered a separation line between the dynamics of the shelf and deep ocean.



A sequence of mesoscale eddies is developed offshore of the western shelf edge with a dominant one
in the Rockall Trough (Fig. 2a and 2b), which are also readily visible in the corresponding SSH pattern
(not shown). The largest amplitudes of the sea level oscillations are observed around the British Isles
and along the southern coasts of the German Bight.

The simulated thermohaline characteristics are consistent with the existing knowledge: the coastal
waters, particularly those in the German Bight, are less saline (Fig. 2c) and represent typical regions of
freshwater influence (ROFIs). In the German Bight, most of the low-salinity water originates from the
Rhine, Ems, Weser and Elbe Rivers and spreads along the Dutch, German and Danish coasts before it
reaches the Skagerrak, where it mixes with the low-salinity outflow from the Baltic Sea. The pattern of
the salinity gradient (Fig. 2d) reveals features along the coasts, in the Norwegian Trench and at the
major fronts in the German Bight and Norwegian Trench.

In the winter, the overall temperature distribution is characterized by cold temperatures on the
shallow shelf and a south-north temperature gradient in the deep water south of Ireland (Fig. 2e). A
warm water plume exits the English Channel and traces the pathway of warm Atlantic water in the
North Sea, which is also known from the satellite observations of Pietrzak et al. (2011). The East Anglia
Plume and the Frisian Front are promptly visible as low-temperature areas originating from the East
Anglia coast (Fig. 2e). The two current branches in the Norwegian Trench associated with the two
opposing flows (one flowing to the east along the southern slope and another flowing in the opposite
direction along the northern coast) are also easily observed as areas characterised by large temperature
gradients (Fig. 2f) coinciding with the salinity gradient maximum; a number of mesoscale features occur
in the deep ocean along the rims of currents (compare with Fig. 2a). In the summer, the warmest
temperatures can be found on the shallow shelf, especially on the Armorican Shelf (Fig. S2a). The
summer temperature distribution is also characterized by well-pronounced temperature gradients. The
simulated gradients along the Celtic Sea Front, Ushant Front, Islay–Malin Head Front and the
Flamborough Head Front (black circles in Fig. S2a) support the results of Pingree and Griffiths (1978).
The disappearance of some of these fronts in the results of the NTE demonstrates that they are tidal



mixing fronts (see Fig. S2a and S2b). Overall, Fig. 2 supports much of what is known from previous
studies (e.g. Pätsch et al., 2017).

Understanding the differential properties of currents is of utmost importance to understand the
propagation of tracers. Therefore, we will present a brief analysis of deformation, as proposed by
Smagorinsky (1963):

$$|D| = \sqrt{D_T{}^2 + D_S{}^2} = \sqrt{\left(\frac{\partial u}{\partial x} - \frac{\partial v}{\partial y}\right)^2 + \left(\frac{\partial u}{\partial y} + \frac{\partial v}{\partial x}\right)^2}$$  (1)

with horizontal tension strain

$$|D_T| = \sqrt{\left(\frac{\partial u}{\partial x} - \frac{\partial v}{\partial y}\right)^2}$$  (2)

and horizontal shearing strain

$$|D_S| = \sqrt{\left(\frac{\partial u}{\partial y} + \frac{\partial v}{\partial x}\right)^2} .$$  (3)

Figure 2g shows the 25-h averaged deformation obtained from the CR surface currents. The order
of this property $O\,(10^{-5})$ is within the ranges measured by Molinari and Kirwan (1975) with Lagrangian
drifters. The most obvious features are the two large areas on the shelf exhibiting low deformation,
namely, the North Sea and the Celtic Sea, including the Armorican Shelf connected by the English
Channel and Southern Bight, where several localized high-deformation areas appear (compare Fig. 2g
with 2b). High-deformation areas are also present in the Irish Sea extending to the northern coast of
Ireland. The difference between the CR and NTE clearly shows the effects of tides on the deformation
in these three areas (Fig. 2h). Despite these shallow, enclosed areas, the deformation along the shelf
edge of the Celtic Sea is also affected by tides. High-deformation features in the deep ocean arise at the
eddy boundaries (compare Fig. 2g with 2a) and are also present in the NTE; hence, flow deformation
is expected to be of significant importance for water masses in the deep ocean. Exceptions are the Bay
of Biscay and the northwest of the domain where the deformation is less pronounced in the NTE. The
difference patterns there have scales of mesoscale eddies suggesting that these eddy dynamics could be
coupled to the one of tides. In the Norwegian Trench, the influence of tides on deformation is strong
due to the small-scale dynamics associated with the two currents along the north and south topography
slopes and the eddies between them.

*2.4 Particle release experiments*

Lagrangian particles are released in the hydrodynamic model, and their propagation is used to analyse
the transport properties. The experiments were carried out "online"; that is, the particle trajectories were
computed within the hydrodynamic model at every time step. Additional experiments were carried out
"offline" using the model velocity output. High-frequency processes and vertical transport are better
accounted for in the former experiments. This intercomparison between the online and offline
integrations demonstrates that neither approach leads to drastic differences when comparing 2-D
particle transport properties.

The online advection of Lagrangian particles was achieved by the freely available open-source

ARIANE model (http://stockage.univ-brest.fr/~grima/Ariane/). The version of ARIANE implemented
in NEMO has frequently been used in other studies, e.g. Blanke and Raynaud (1997) and Blanke et al.
(1999). Further details of the ARIANE model can be found in the appendix of Blanke and Raynaud
(1997) and in the ARIANE user manual. Beaching is not possible; that is, the total number of particles
remains constant over time. An extra wind drag is not used, nor is additional horizontal diffusion for
the particles, because it is assumed that the high temporal resolution of the velocity fields and large
velocity gradients will provide a sufficiently high diffusivity (van Sebille et al., 2018). The vertical
velocity is taken into account, and the particle positions are written hourly.



Different seeding strategies were implemented. In one class of experiments (#1 and 4–6 in Table
2), the particles were seeded at 1 m (surface particles), as well as in the grid cells just above the seafloor
(bottom particles). In the second experiment (#2 in Table 2) named CR-V, particles were released in a
100 km wide stripe extending oceanward from the 150 m isobath starting in the Bay of Biscay and
ending north of the Shetland Islands at 61.7° N; in this experiment, particles were seeded vertically
every 20 m. In a third experiment (#3 in Table 2) named CR-B, the particle tracking process was carried
out offline; for this purpose, the freely available open-source model OpenDrift (Dagestad et al., 2017)
was used, in which the particles were advected by a 2$^{nd}$-order Runge–Kutta scheme. The offline
calculation was performed backward in time at a constant depth with a velocity input time step, a model
time step and an output time step of 1 h. The particle release depth in this experiment was 1 m.

The seeding strategy was consistently executed as follows. The initial distribution of particles was
uniform with 1 particle per model grid cell, that is, 64,831 particles per depth layer for the whole domain
(experiments #1 and 3–6 in Table 2) and a total of 345,011 particles in experiment #2. In the CR and
CR-V (experiments #1 and 2, respectively), particle release was repeated on the first day of every month
in 2015, and particles were traced for 1 month. Thus, 12 data sets, each including 1 month of trajectory
data, were generated. Additionally, for the seeding in January, the particle positions were saved for 6
months. The CR-B, FWE and NWE (experiments #3, 5 and 6, respectively) were conducted only for
January 2015; the NTE (experiment #4) was further performed for July 2015.

**Table 2**: Summary of the particle release experiments; further details are given in Sect. 2.4.

| # (abbr.) | Particle advection | | Spatial seeding | | | Integration time | Details |
|---|---|---|---|---|---|---|---|
| | Online | Offline | Whole domain | Shelf edge | Vertical | | |
| **1 (CR)** | x | | x | | surface & bottom | 12 x 1 month | 3-D particle motion |
| **2 (CR-V)** | x | | | x | every 20 m | 12 x 1 month | 3-D particle motion |
| **3 (CR-B)** | | x | x | | surface | January | backtracking |
| **4 (NTE)** | x | | x | | surface & bottom | January + July | no tides |
| **5 (FWE)** | x | | x | | surface & bottom | January | filtered wind |
| **6 (NWE)** | x | | x | | surface & bottom | January | no wind |





*2.5 Particle density trend*

The analyses of the results will focus on typical Lagrangian properties, e.g. the positions of the particles
and their trajectories. Such a presentation could be considered inferior compared with the Eulerian
presentation, which displays the concentrations of properties. However, from these Lagrangian
characteristics, one can derive properties similar to the concentration that can represent the
"compaction" process of particles in certain areas or identify the areas that are more frequently visited
by the Lagrangian particles. These properties related to particle density allow the areas in which
particles accumulate to be identified.

Different approaches to quantify particle accumulation have been proposed (Koszalka and LaCasce;

2010; Koszalka et al., 2011; van Sebille et al., 2012; Huntley et al., 2015). Below, in addition to the
typical Lagrangian properties, a property named the "density trend (DT)" is introduced that measures
the number of particles that have visited each grid cell during a certain time interval. This quantity is
normalised by the corresponding number of particles in the motionless situation for the same time
interval, in our case, 1 particle per grid cell:

$DT(x, y, t_n) = \dfrac{\sum_{i=0}^{n} N_{u \neq 0}(x, y, t_i)}{\sum_{i=0}^{n} N_{u=0}(x, y, t_i)}$                      (4)

where $DT$ is the density trend, $(x, y)$ are the coordinates of an arbitrary grid cell with dimensions ($dx$,
$dy$), $n$ is the number of time steps from $t_0$ to $t_n$, $u$ is the velocity field and $N$ is the number of particles at
time step $i$ in grid $(x, y)$. In the present study, ($dx$, $dy$) represent the model grid dimensions but could be
larger or smaller for other applications. A DT greater (smaller) than unity corresponds to more (fewer)
particles, which are identified in a grid cell on average, than there would be without currents. Thus, the
DT can be interpreted as the percentage of the initial number of particles averaged over time.

The definition of the DT is not straightforward if the initial particle concentration is zero in some

areas. If the number of particles in some areas remains small (e.g. areas close to an inflow-dominated



open boundary or divergence zones), the statistical confidence of this property cannot be ensured.
Therefore, areas where the DT is less than 30 % are excluded from the analysis (white areas in the
following figures). Overall, for integration times longer than 1 month, large areas remain free of
particles; thus, $t_n$ is chosen to be 1 month in the present study.

**3 Results and discussion**
*3.1 Overall analysis of trajectories and particle dynamics*
The particle trajectories (Fig. 3) of the CR (experiment #1, see Table 2) show the well-known, dynamic
features of the ENWS and the surrounding deep ocean. In relatively shallow areas, e.g. the English
Channel, Southern Bight and Irish Sea, the surface and bottom currents reveal similar patterns, which
is typical for wind-driven shallow water circulation (Fig. 3a and 3b). Trajectories symbolising currents
appear relatively thick in the areas dominated by strong tides because the large-scale presentation cannot
effectively resolve small tidal excursions. This is supported by the magnified representation of the
dynamics in Fig. 3c and 3d. After 12 h, the trajectories on the shelf present as nearly closed circles. The
difference between the start and end positions on the circular loops denotes the net transport, which is
much smaller than the tidal excursions. The net transport rapidly increases, and the tidal excursions
decrease further off-shelf beyond the 900 m isobaths, where the mesoscale dynamics are dominant. As
in the case of the Eulerian visualisation of the velocity field, the 200 m isobath can be considered the
boundary separating the dynamics of the shallow and deep ocean. The meandering of the European
Slope Current along the shelf edge (at ~500–2,000 m) is pronounced from the Bay of Biscay to the
Goban Spur and around the Porcupine Bank (Fig. 3b). The Skagerrak and Norwegian Trench also show
pronounced mesoscale dynamics (Fig. 3d).

*3.2 Tendencies of particle accumulation*
*3.2.1 Surface and bottom patterns of the particle distribution*
Despite some similarities between the surface and bottom trajectories (Fig. 3), the particle accumulation
patterns in shallow areas are considerably different. To investigate these differences, the positions of
the particles released in January (CR, experiment #1, see Table 2) are displayed in Fig. 4. In the
following, "+" and "-" symbols will be used to denote locations of particle accumulation ("+") and
removal ("-"). After 1 month, the surface-released particles accumulate mainly along narrow patterns
on the shelf and in the Skagerrak (Fig. 4a). In contrast, the coastal regions around Great Britain and
Ireland (but also in the German Bight) can be considered divergence zones. The particle distribution in
the deep ocean also shows small stripe-type patterns, especially in the southwestern part of the model
domain.

There is a tendency for the bottom-released particles to leave areas with a steep bottom slope. The

most obvious example is the continental slope are along the 200 m isobath from the Spanish coast
around the Goban Spur and Porcupine Bank (Fig. 4b) until the Norwegian Trench. Van Aken (2001),
Huthnance et al. (2009) and Guihou et al. (2018) demonstrated that slope currents, downward flows of
shelf water and breaking internal waves, respectively, dominate the dynamics of the ENWS continental
slope. Thus, this tendency of bottom-released particles to leave the continental slope is consistent with
the findings of earlier studies on the dynamics of the ENWS shelf edge.

After 6 months, vast areas of the shelf and the western part of the domain become free of particles.

The Lagrangian particles flow from the English Channel along the Frisian Front in the south and the
Fair Isle Current in the north into the inner North Sea (Fig. 4c). The pattern in the Irish Sea is similar to
the Frisian Front: a narrow stripe of particles in the middle of this basin is the remnant of a similar stripe
from an earlier time (Fig. 4a) connecting the source of particles (in the south) to their sink (in the north).
The region around the Orkney and Shetland Islands accumulates particles owing to on-shelf transport
by the westerlies. This region additionally receives particles from the south originating from the Irish
Sea or having flown around the western coast of Ireland; in both cases, these particles are sourced from
the deep ocean. Bottom accumulation on the shelf occurs mainly south of the Dogger Bank (Fig. 4d).
Also the bottom trajectories (Fig. 3b) show that particles north of Dogger Bank are forced to flow
around its western edge through a narrow channel into the basin to its southeast (see the bathymetry in
Fig. 1) suggesting topographically influenced particle motions. Once the particles reach this basin, they





can flow out only northward along its thalweg until they reach the northeastern edge of Dogger Bank
(compare Fig. 4d with Fig. 1).

In the Skagerrak, the situation is as follows. At the bottom, the Norwegian Trench supplies the

Skagerrak with particles from the Atlantic along its southern slope. At the surface, the Skagerrak
receives particles from the German Bight and the Baltic Sea. Particles approaching the Skagerrak can
become trapped in its circular and eddy-dominated velocity pattern (Fig. 3d and 2b), which extends
from the surface down to the bottom (see also Rodhe, 1987; Gutow et al., 2018). In the Norwegian
Trench, the particle distribution is ambiguous due to the irregular mesoscale dynamics therein.

The current particle positions are not sufficiently representative of their accumulation and dispersal

over long periods and can lead to misinterpretations of their accumulation trends. This becomes evident
by comparing the particle positions after 1 month (Fig. 4a and 4b) with the DT for the same period (Fig.
5a and 5b). Although the general surface and bottom patterns (Fig. 4a and 4b) for January 2015 are
comparable to the mean accumulation patterns (Fig. 5a and 5b), some features do not coincide. The
monthly DTs for all twelve months are shown in Figure S3 and S4 for the surface- and bottom-released
particles, respectively. At the surface, only a few accumulation areas are visible on the annual mean
map (red areas in Fig. 5c); these areas are located in the Irish Sea (*1*), English Channel (*2*), Southern
Bight (*3*), German Bight (*4*), Skagerrak (*5*), along the continental slope (*6*), and at the Fair Isle Current
(*7*). Vast coastal areas have a DT smaller than 0.3, implying off-coast propagation. Despite the
numbered accumulation areas and coasts prone to particle removal, most of the domain shows neither
particle accumulation nor removal (DT $\approx$ 1).

At the bottom, particle accumulation is rather ambiguous in the deep ocean, but the removal of

particles from areas with steep topography is evident (Fig. 5d). On the shelf, the tendency of particles
to propagate away from coasts is reduced; particles even accumulate, e.g. in the German Bight and
along the eastern British coast (discussed in detail in Sect. 3.3). Further, accumulation takes place to



the south of Dogger Bank and in the Skagerrak. It is worth noting that the major accumulation pattern
at the surface along the Frisian Front (*3*, *4*) has as its counterpart a pattern of removal at the bottom
(compare Fig. 5c and Fig. 5d). Additionally, coastward of accumulation area *4*, there is a removal area
at the surface; in the same area, the bottom pattern shows a tendency of accumulation. These opposite
tendencies in the surface and bottom layers suggest that the vertical circulation is also important in
shallow environments. The inflow along the southern slope of the Norwegian Trench appears as
increased particle accumulation.

There are some prominent small-scale features (stripe-like or filament-like characteristics)
occurring in the deep ocean (Fig. 5a and S3). These features change their positions depending on the
eddy motion. They are reminiscent of the attributes reported by Haller and Yuan (2000), who
demonstrated that particles initially located outside eddies accumulate in lines along the boundaries
between them. When the averages are computed for a longer period, these filaments tend to disappear
(compare Fig. 5a with 5c), which is explained by the fact that the time scales of eddy motions are
substantially shorter than the annual scale. This follows from the changes in the position and occurrence
of the stripe-type areas with DTs greater than 1 from month to month (compare the results from single
months of Fig. S3). A more profound Lagrangian representation of eddies and their coherent character
can be found in, e.g. Beron-Vera et al. (2018).

*3.2.2 The role of tides*
The difference in the January DTs between the CR and NTE (experiments #1 and 4, respectively, see
Table 2 and Fig. 6a and 6b) demonstrates that the tidal forcing considerably affects the accumulation
patterns on the shelf. The largest differences between the two experiments (marked with "***" symbols)
occur along the Frisian Front and along the front in the Irish Sea (Fig. 6a and 6b). These differences
show greater accumulation at the surface in the CR at the western flanks of these fronts and less
accumulation at the eastern flanks than in the NTE. Obviously, the tidal signal affects frontal-like
structures. Similar differences between the two experiments appear in the English Channel, around the





north of Great Britain, and at the continental slope of the Celtic Sea. In most of the remaining parts of
the domain, these differences are rather small.

Nevertheless, tides also affect the accumulation of particles in the deep ocean, which is dominated
by sub-basin-scale eddies, as well as other areas in the Bay of Biscay and in the Norwegian
Trench/Skagerrak, which are dominated by mesoscale motions. This could serve as another indication
of the interaction between tides and mesoscale dynamics.

The most pronounced large-scale feature in the differences observed at the sea surface is in the
vicinity of the shelf (Fig. 6a). At the bottom (Fig. 6b), the largest differences between the two
experiments occur beyond the 200 m isobaths in the direction of the open ocean. The changing sign of
the difference reflects large oscillations at small scales, possibly indicating that a further increase in the
model resolution is needed to adequately resolve the accumulation and dispersion of particles in the
area of the continental slope. Bottom patterns in the North Sea are also present and clearly demonstrate
the importance of tides as a driver of particle accumulation. The principal patterns are similar to the
surface: around Great Britain the difference signal is rather ambiguous; in the Southern Bight and
southern North Sea the difference patterns are rather distinct and follow the flanks of the East Anglia
Plume and Dogger Bank. In the front itself, the tides do not have a significant impact on the
accumulation of particles. This comparison between the surface and bottom patterns indicates that,
unlike the currents, which do not drastically change in the vertical direction in the shallow ocean, the
accumulation of particles at the bottom is different from that at the sea surface. This suggests that the
current shear affected by tides modifies the particle accumulation patterns. This finding is supported by
the influence of tides on deformation (compare Fig. 6a and 2h), the pattern of which partly coincides
with the DT difference.

*3.2.3 The role of wind*
A large part of the variability is caused by atmospheric variability (mostly on synoptic time scales)
(Jacob and Stanev, 2017); therefore, we will analyse the contributions of wind to the accumulation and



dispersion of particles in the FEW and NWE (experiments #5 and 6, respectively, see Table 2). It is
worth noting that the ranges of the responses to wind are comparable to the responses to tides. The
overall conclusion from the comparison among the differences in the surface properties between the
CR and NTE (Fig. 6a) from one perspective and between the CR and FWE (Fig. 6c) from another
perspective is that the largest differences caused by tides and winds occur in almost the same areas: the
Frisian and Irish Sea Fronts, the continental slope, and the Norwegian Trench; the English Channel is
less influenced in FWE. Smoothing the wind (FWE) makes the accumulation stripes "sharper", whereas
the short-term wind forcing tends to "blur" the particle distribution. However, turning the wind off
(NWE) changes the accumulation patterns significantly (compare Fig. 6e and 6c). The most affected
areas are (1) the coastal areas of Great Britain and Ireland, (2) the Skagerrak, which no longer
accumulates particles, (3) the mouths of the Rhine and Elbe rivers which extend further to the west, and
(4) the accumulation area along the Celtic Sea shelf edge that disappears in the NWE. Reducing the
variability of the wind or turning it off completely also has very pronounced impacts on the bottom
particles (compare Fig. 6d and 6f). The accumulation patterns at the bottom are mostly affected in the
northern part of the shelf and the Norwegian Trench/Skagerrak.

The difference between the FEW and NEW (not illustrated here) demonstrates that, on the shelf,

the westerlies are essential for particle accumulation.

*3.2.4 The role of fronts*
The high-salinity and high-temperature gradients (fronts) in Fig. 2d and 2f are similar to the DT patterns
shown in Fig. 5a. These fronts support the ones reported by Belkin et al. (2009), particularly the fronts
in the southern North Sea. Additionally, in terms of the yearly averaged DT (Fig. 5c), the DT maxima
coincide with the known front positions; in contrast, not all detected fronts show particle accumulation.
There are also some differences from the analysis of Pietrzak et al. (2011), who analysed the dynamics
of the Frisian Front and East Anglia Plume using satellite data of the sea surface temperature (SST) and
suspended particulate matter (SPM). The differences between the present simulations and the results of

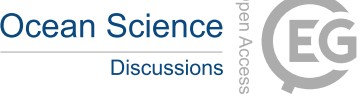



Pietrzak et al. (2011) in the East Anglia Plume are mostly because the particles in the model have a
neutral buoyancy and because no particle sources are prescribed (the seeding is uniform).

To demonstrate the ability of a front to accumulate particles, a surface section across the Rhine

Plume (Frisian Front) is chosen as an example (solid black line in Fig. 5a; see also its inset). The front
separates the waters of the English Channel  (higher salinity) and the Rhine ROFI (lower salinity). In
Fig. 7, the graphs start in the west (left) and end in the east (right). The maximum DT in the CR (left
vertical dashed line) is located where the salinity and temperature start to decrease (~34.5 to 28.7 PSU
and ~12.7 to 11.1°C, respectively). The related density changes are ~4.51 and ~0.30 kg m$^{-3}$. Hence, the
salinity causes the density gradient which in turn influences the accumulation of particles. In the
backward simulation (CR-B), the DT maximum (right vertical dashed line; experiment #3, see Table 2)
is at the same location with respect to the salinity change when the particles are coming from the
opposite direction (forward: southwest, backward: northeast). The peaks of the DT curves are bounded
by a rather constant DT, which is higher on the side of particle supply than on the side of particle
dispersion in CR. In CR-B the particle supply is hampered by the front. These results are very similar
to what has been suggested by Lohmann and Belkin (2014) (see their Fig. 2). Despite the vertical
dynamics (see Sect. 3.3), particle accumulation along fronts can be explained considering that the mean
velocity is not parallel to the front but oriented further clockwise (in the CR the orientation of the front
is almost in the north–south direction, whereas U is veered clockwise). This would lead to a crossing
of the front by particles, but the particles are hindered by the (haline) front and flow along it. In the CR-
B, the dynamics are reversed, and thus, particles accumulate on the other side of the front. Particle
accumulation along other ROFIs can also be observed at, e.g. the western Danish coast along the Elbe
River outflow. Postma (1984) called the boundary of the Wadden Sea a "line of no return" whose
location is comparable to a strong salinity gradient (Fig. 2c and 2d). From the results of the present
study, this interpretation of the boundary of the Wadden Sea can be confirmed: if a particle of the
German Bight crosses the front, it is unlikely that it will be able to return.

In the NWE (Fig. 6e), the Frisian and Irish Sea Fronts are less pronounced than in the CR,
demonstrating the intensification of frontal accumulation by wind. Due to the missing westerly wind,
particles are no longer transported to the fronts, where they can accumulate in the areas of thermohaline
gradients. This is especially true for regions where the wind is constantly blowing in the same direction,
e.g. regions within the westerlies.

Another kind of shelf front is a tidal mixing front (Sect. 2.3 and Fig. S2), whose dynamics have
been described repeatedly (e.g. Hill et al., 1993). These fronts are known to accumulate natural and
artificial flotsam (Simpson and Pingree, 1978). Analyses of the January NTE results do not reveal that
the DT maxima disappear if the tides are turned off. In July, tidal mixing fronts are clearly visible as
temperature gradients (Fig. S2a), and some of them can be observed in terms of DT patterns (Fig. S2c).
In contrast to January, these fronts disappear in July if the tides are turned off (Fig. S2d). Due to their
seasonal occurrence, these tidal mixing fronts are less pronounced in the yearly averaged DT then others
are, e.g. the fronts of ROFIs. However, not all tidal mixing fronts occur as DT maxima. Although the
well-known jet-like velocities along fronts can be seen in U in Fig. 7, the horizontal model resolution
is possibly too coarse; that is, the model cannot resolve all important frontal dynamics.

*3.3. Vertical circulation in the North Sea*
Although shelf dynamics are dominated by strong horizontal motions, they cannot be considered fully
two-dimensional. Examples of the role of vertical processes are given by tidal mixing fronts (Garret
and Loder, 1981; van Aken et al, 1987), upwelling in the German Bight (Krause et al., 1986), tidal
straining (de Boer et al., 2009) and secondary circulation in estuaries. The differences between the
surface and bottom accumulation patterns described in Sect. 3.2.4 (Fig. 5a and 5b) are indicative of the
role of vertical processes. Such indications are clearly observed in the map of the differences between
the vertical positions of particles released at the bottom after one month of integration (Fig. 8a). These
differences are very pronounced along the eastern British coast, eastern Irish coast, around the Dogger
Bank and in several smaller coastal areas, e.g. at the western French coast. Some of these patterns are
topographically induced, like the one at the Dogger Bank, where particles from the northwest ascend



and particles released on the Dogger Bank descend. However, a DT at the surface smaller than 1 and a
DT at the bottom greater than 1 (compare Fig. 5c with 5d) suggest an upward movement of water.
Single particle trajectories along the British coast reveal that the bottom flow is directed coastward and
offshore at the surface (small inset in Fig. 8a). Locations with dominant upwelling or downwelling are
indicated with "↑" and "↓" symbols, respectively, in Fig. 8a.

In the backtracking experiment (CR-B, experiment #3, see Table 2; Fig. 8b), particles accumulate

in coastal upwelling areas, emphasising the dynamics described above. The opposite situation is present
on the western Irish coast and in the western Irish Sea; here, a downward movement of water can be
observed. The NWE shows that the main driver of coastal water transport at meridionally oriented
coasts is the prevailing westerlies (Fig. 6e and 6f). Without wind, the eastern Irish and British coasts
have DT values clearly exceeding 0.3; with the original wind forcing, these areas have DT values
smaller than 0.3. In contrast, the DT of the western Irish coast and in the western Irish Sea is reduced
in the NWE. These results also support the theory of Lentz and Fewings (2012) regarding wind-driven
inner-shelf circulation.

Wind forcing is not the only explanation for the offshore-directed transport at some of the shelf

coasts, particularly along the eastern British coast. Downwelling at fronts is associated with upwelling
on the coastward side as a result of coastward transport at the bottom and offshore-directed transport at
the surface. Similar effects have been modelled (Garret and Loder, 1981) and observed (van Aken et
al., 1987) in previous investigations.

*3.4. Dynamics at the shelf edge*

The analysis below uses the results of the CR-V with the seeding prescribed in a 100 km wide segment
extending oceanward from the 150 m isobaths (experiment #2, see Table 2). The exchange of particles
between the deep ocean and the shelf is estimated by the number of particles crossing the 200 m isobaths
and the changes in their depth with respect to the depth at which they were released. The 100 km wide
segment is divided vertically into four parts: from the surface to 80 m (Fig. 9a), 100–180 m (Fig. 9b),



200–280 m (Fig. 9c) and 900–980 m (Fig. 9d). The major result of this experiment is that with increasing
depth (1) the dispersion of the cloud of particles in the vicinity of the 200 m isobaths decreases, and (2)
particles in the deeper layers do not penetrate onto the shelf. Many particles released above 100 m move
onto the shelf; their depth remains almost unchanged or even decreases (Fig. 9a). In the three deeper
intervals of release, deep oceanward transport is dominant (red stripe along the 200 m isobath in Fig.
9b, c and d). These dynamics, which are sketched in Fig. 9e, support the results of Holt et al. (2009),
Huthnance et al. (2009) and Graham et al (2018), whose simulations also showed shelfward transport
distinctive of the upper 150 m along the 200 m isobath; below 150 m, they found deep oceanward
transport. The simulated exchanges between the shelf and open ocean (the extent and direction of
particle propagation) are also in overall agreement with the recent results of Marsh et al. (2017), who
analysed drifter observations and Lagrangian simulations at the ENWS continental slope. Down to 280
m, particles propagating away from the continental slope form filaments or eddy-like patterns as in the
Rockall Trough; another fraction of the particles are advected within the slope current. Although the
latter are covered by the coloured areas, some of them are visible at the entrance of the Norwegian
Trench. The underlying dynamics at the ENWS continental slope are discussed in Sect. 3.2.1.

**4 Conclusions**
Lagrangian analyses in conjunction with Eulerian analyses revealed physically distinct regimes in
different parts of the study area. The underlying dynamics were investigated in terms of particle
accumulation and removal, which were quantified by the DT quantity.

•    On the shelf: Fronts act as barriers and accumulate particles. Tides affect the positions and

appearance of particle accumulation in frontal areas. Vertical water transport at meridionally

oriented coasts on the shelf is influenced by westerlies. Offshore-directed wind induces a DT

smaller than 1; the situation is reversed for onshore-directed wind.

•    In the deep ocean: Eddies influence the particle dynamics on short time scales (individual

565         months); however, an annual mean DT ≈ 1 reveals an absence of long-term stable accumulation

areas. Tides affect the DT, suggesting the interaction of tides and mesoscale dynamics.





- •    Shelf edge (200 m isobath): The shelf edge represents a transition zone from the wind- and
- tidally driven shallow shelf regime to a baroclinic eddy-dominated deep ocean regime. The
- shelf edge shows on-shelf transport in the upper layers and downwelling-like off-shelf-directed
- transport below 100 m. Bottom current branches tend to remove particles from the continental
- slope.
- •    At the surface: Accumulation patterns on the shelf show high variability on monthly time
- scales; some accumulation areas remain stable on yearly time scales. These long-term stable
- zones occur mainly along the fronts of ROFIs and in the Skagerrak. At the shelf edge, particles
- are transported onto the shelf by westerlies. The influence of wind on particle accumulation is
- on the order of the influence of tides.
- •    At the bottom: On the shelf, bottom currents are mainly influenced by the topography and
- follow its thalweg.


The differences in the properties of the velocity field (e.g. deformation) reveal two different
regimes: a shelf regime with rather little deformation and a deep ocean regime with considerable
deformation. On the shelf, tidally induced deformation plays a substantial role in particle accumulation
and dispersal.

The present study demonstrates the illustrative potential of Lagrangian methods. In conjunction
with traditional Eulerian analysis, Lagrangian analysis can enhance the interpretation of observed or
simulated dynamics and provide a solid basis for estimating the propagation of floating marine debris.

*Code/data availability*: The model codes of NEMO, ARIANE and OpenDrift as well as the GPS
drifter and HF radar data are freely available. Scripts and data can be obtained by a request to the
corresponding author.



*Author contributions.* MR and EVS conceived the study. MR performed the model runs and analysed
and prepared the figures. MR and EVS interpreted the results, and MR prepared the manuscript with a
significant contribution from EVS.

*Competing interests.* The authors declare that they have no conflicts of interest.

*Acknowledgments.* We thank the UK Met Office and Joanna Staneva for providing the NEMO AMM7
setup. This study was carried out within the project "Macroplastics Pollution in the Southern North
Sea – Sources, Pathways and Abatement Strategies" (grant no. ZN3176) funded by the German
Federal State of Lower Saxony. The authors thank Sebastian Grayek for technical support and Jens
Meyerjürgens for carefully reading the manuscript and giving important advice.

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



**Figure captions:**
**Fig. 1**. Bathymetry of the model domain. The shelf is defined as depths shallower than 200 m (colour
bar within the map). In this and all following figures, the 200 m depth contour is highlighted with a
black solid line. The abbreviations used in the text are as follows:
Armorican Shelf (*AS*), Bay of Biscay (*BB*), Celtic Sea (*CS*), Dogger Bank (*DB*), East Anglia (*EA*),
English Channel (*EC*), German Bight (*GB*), Goban Spur (*GS*), Irish Sea (*IS*), Kattegat (*Ka*), North Sea
(*NS*), Norwegian Trench (*NT*), Rockall Trough (*RT*), Skagerrak (*Sk*), Southern Bight (*SB*), St.
George's Channel (*SGC*), Orkney/Shetland Islands (*OI/SI*), Outer Hebrides (*OH*), Porcupine Bank
(*PB*), Fair Isle Current (*a*), European Slope Current (*b*), East Anglia Plume (*c*), Frisian Front (*d*),
Rhine River (*1*), Ems River (*2*), Weser River (*3*) and Elbe River (*4*).

**Fig. 2.** Simulated surface properties in the CR for January 2015: velocity magnitude derived from the
averaged u und v velocity components (a), mean velocity magnitude (b), mean salinity (c) and mean
temperature (e). Magnitudes of the temperature (d) and salinity (f) gradients as well as the
deformation (g) in the CR and the differences between the deformation in the CR and NTE (h) are
presented as 25-h averaged fields on 15 January 2015.

**Fig. 3**. Lagrangian trajectories of every 5$^{th}$ particle after 15 days of integration released on 01 January
2015. Particles are released at the surface (a) and bottom (b) (experiment #1, see Table 2). (c) and (d)
are magnified views of the domain showing the trajectories of the first 12 h (c) and 24 h (d)
representative of different dynamics: the area of the Armorican Shelf continental slope including tidal
ellipses (c) and the circulation in the Skagerrak (d). The trajectory colours are randomly chosen for
better visibility. Grey lines are isobaths in 700 m (c) and 400 m steps (d).

**Fig. 4**. Particle positions after 1 month (a) and (b) and 6 months (c) and (d) released on 01 January
2015 of surface-released (a) and (c) and bottom-released particles (b) and (d) (experiment #1, see
Table 2). "+" and "-" symbols represent areas with pronounced particle accumulation and dispersion,
respectively; details are given in the text.




**Fig. 5.** Tendencies of accumulation shown as the January mean DT (a) and (b) and the annual mean

DT for 2015 (c) and (d) (averages of Fig. S3 and S4, respectively). (a) and (c) correspond to surface-

released particles, while (b) and (d) correspond to bottom-released particles (experiment #1, see Table

2). In (a), the solid black line located in the German Bight is the transect shown in Fig. 7 (enlarged in

the inset). The numbers in (c) indicate the most pronounced accumulation areas.

869

**Fig. 6.** Analysis of the sensitivity experiments with respect to tides and wind. (a) and (b) are the

differences between the CR and NTE (experiment #4, see Table 2) in January 2015 at the surface (a)

and bottom (b). (c) and (d) are the corresponding differences between the CR and FEW; (e) and (f) are

the differences between the CR and NWE. The "*" symbols in (a), (b) and (e) denote pronounced

differences; see the text for details.

875

**Fig. 7**. Density trend (DT), salinity (S), temperature (T) and velocity vectors (U) at the surface as the

means of January 2015 along the transect in the German Bight (solid black line in Fig. 5a). The

vertical dotted lines mark the DT maxima of the forward (left one, solid DT line) and backward (right

one, dashed DT line) simulations (experiments #1 and 3, respectively, see Table 2).

880

**Fig. 8.** Difference of the final depth minus the initial depth of bottom-released particles after 1 month

(January) in 2015 (a) (experiment #1, see Table 2). Positive/negative values indicate a depth

increase/decrease. The model grid in which a particle was released is coloured depending on its depth

change. The small figure shows a magnified view of the British coast with two exemplary bottom

(black) and surface (red) trajectories starting at the big dots. The trajectories are detided with a 25-h

flowing mean. The "↑" and "↓" symbols denote coastal areas favourable for upward and downward

water movements, respectively. Tendencies of accumulation are shown as the January mean DT

calculated from the backward simulation CR-B (b) (experiment #3, see Table 2).






**Fig. 9**. Particle positions (purple dots) and particle depth differences with respect to the depth of
release in different depth layers (colours) after 1 month (January) in 2015 computed in the CR-V
(experiment #2, see Table 2). Details of the seeding strategy can be found in Sect. 2.3. The colour
coding shows the difference of the final depth minus the initial particle depth, computed as the mean
difference of all particle depths seeded at the same location in the horizontal plane. All particles
released in the respective depth range are taken into account. Particles were released in four depth
layers: from the surface to 80 m (a), 100–180 m (b), 200–280 m (c) and 900–980 m (d). The dynamics
at the continental slope concluded from (a) to (d) are sketched in (e).



**Figures:**

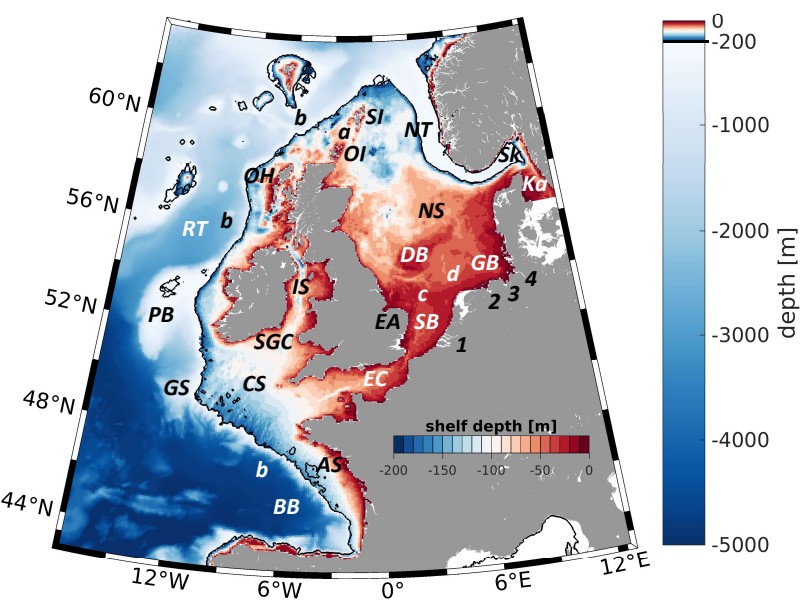

**Fig. 1.**





**Fig. 2.**



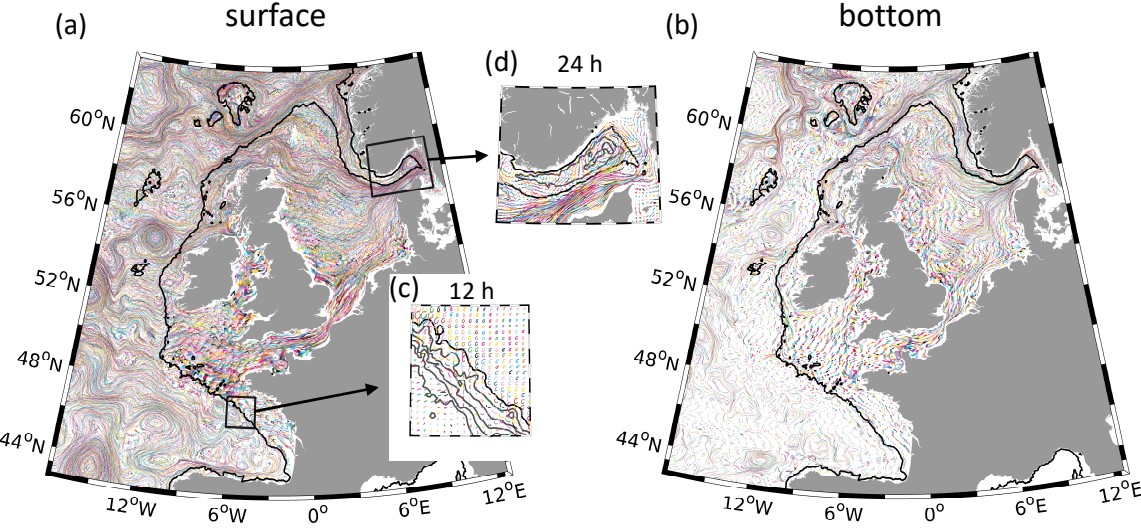

**Fig. 3.**



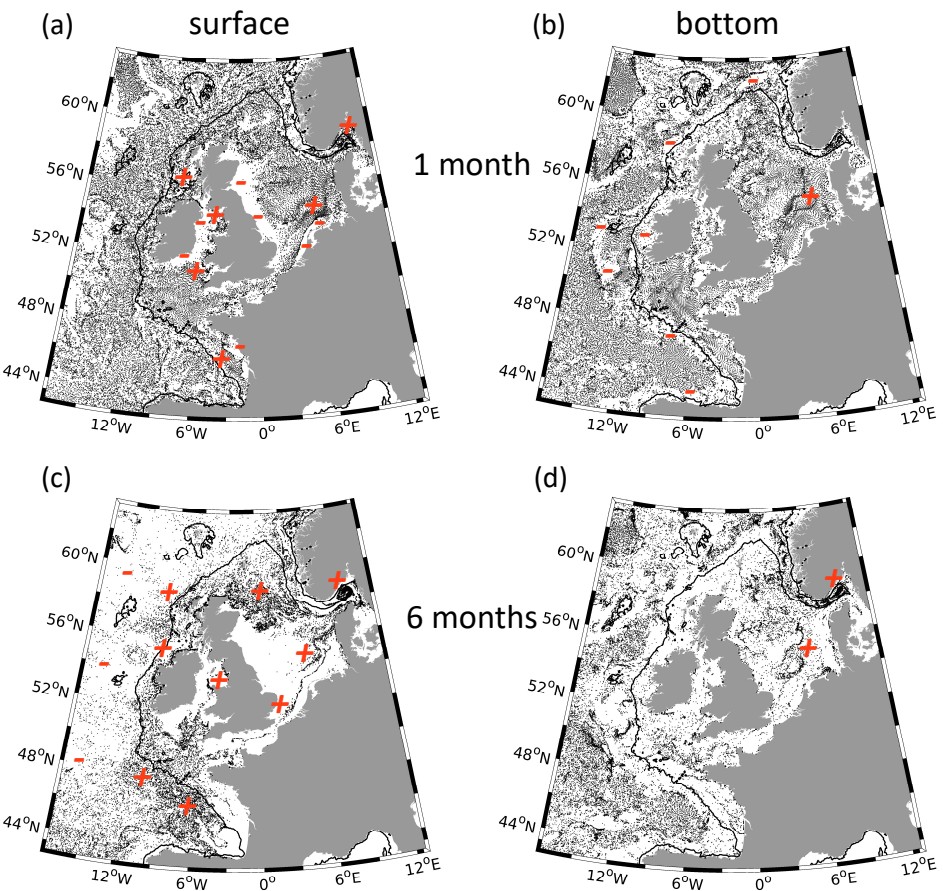

**Fig. 4.**



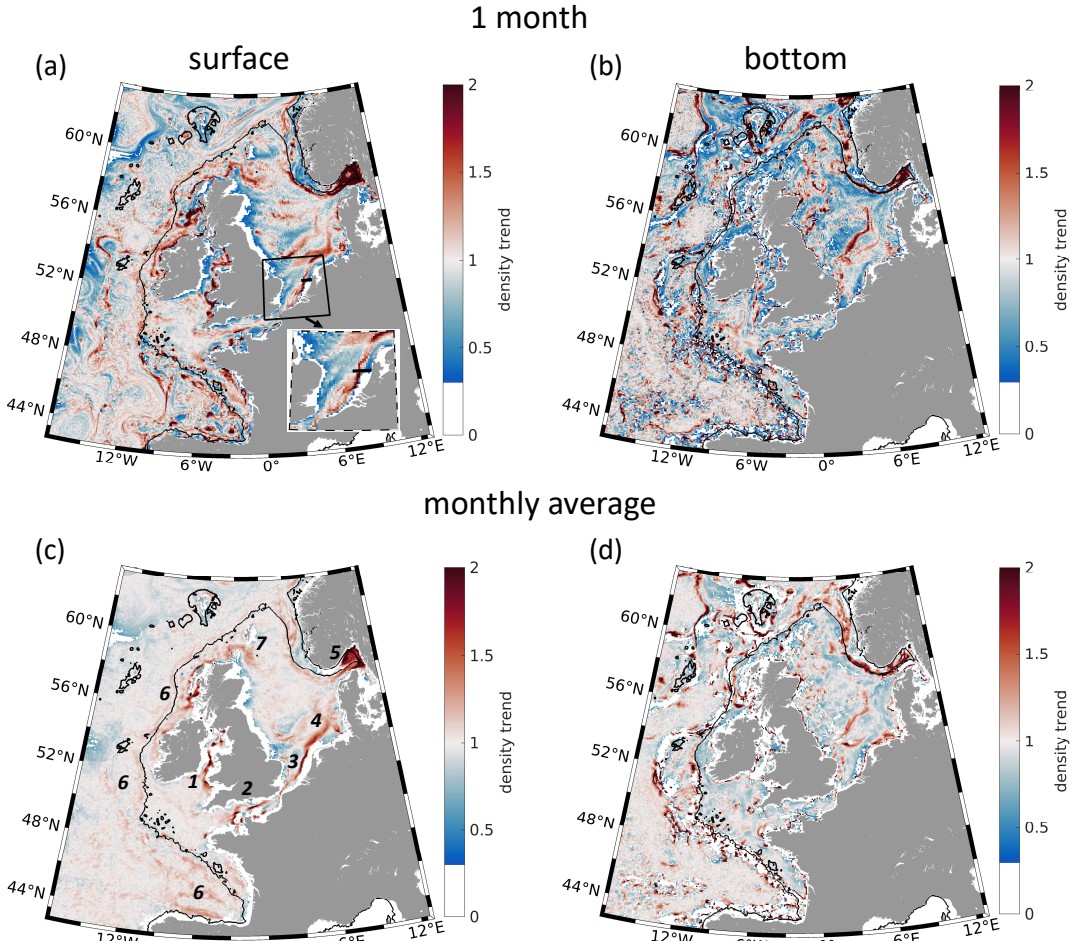

**Fig. 5.**





**Fig. 6.**





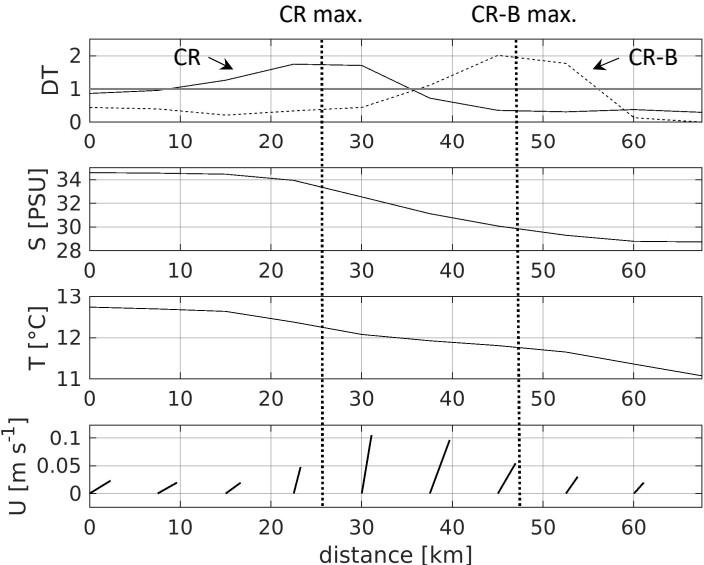

**Fig. 7.**




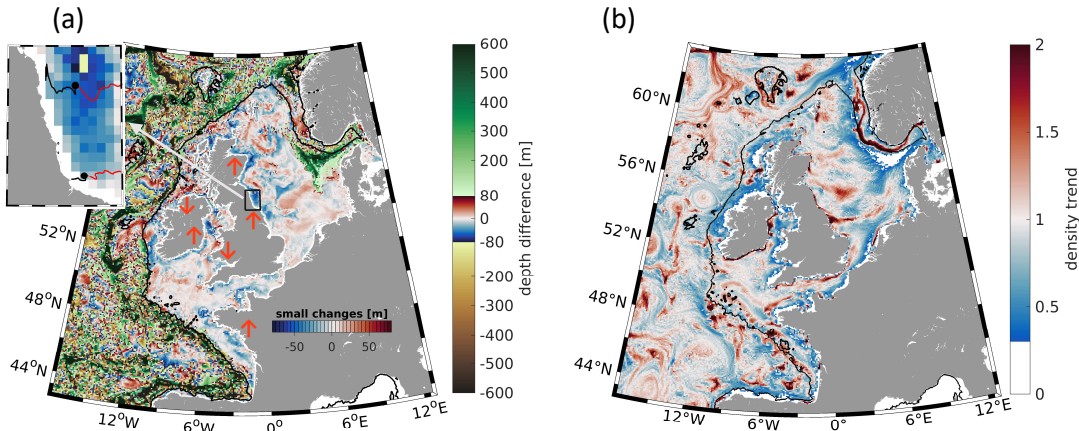

**Fig. 8.**





**Fig. 9.**