# Peer review of "Circulation of the European Northwest Shelf: A Lagrangian perspective"

_Ocean Science, 2019_

## Referee Comment (RC1) · Anonymous Referee #1 · 16 Jan 2020

Review of 'Circulation of the European Northwest Shelf: A Lagrangian perspective' by Marcel Ricker and Emil Stanev.

Summary

The manuscript describes a series of Lagrangian particle tracking experiments carried out to characterise the water circulation and the accumulation of hypothetical particles on the northwest European continental shelf. Particles were released at the surface and the bottom at the start of monthly runs for six different scenarios, mostly uniformly distributed over the area. Some scenarios were carried out over one year, others only for January. A property called 'density trend' is defined to analyse the resulting particle movements. The analysis consists mainly of differences of this 'density trend' between the model scenarios.

[Figure]

General comments

This is an interesting manuscript, with powerful visualisations, supported to a large extent by the newly introduced 'density trend'. However, after reading it, I am left mostly confused, because important information is not supplied (or has escaped my attention), and choices are not motivated (while some of these choices are bound to affect the results). Moreover, the results and discussion section is rather long-winded, depends on visual comparison between figures (which will end up on different pages), and seems to jump unpredictably between figures in quite a few places. It is not clear to me how the particles (are allowed to) move in the vertical (vertical mixing seems to be absent or under-represented). For a number of figures, it is not clear if surface, bottom or e.g. depth-averaged values are presented. Also, I have doubts about the January temperature fields presented/used: the temperatures seem too high, and I am surprised by the magnitude of the spatial gradients in the North Sea. Validation of these is absent. Overall, I would recommend major revisions. I will provide more detailed comments below.

Vertical particle motion

Under well-mixed conditions, as may be expected in most of the North Sea in winter, I would expect neutrally buoyant particles (as simulated here?) to be mixed quickly in the vertical by turbulent mixing processes. That means that there should be little difference in the overall dispersal of surface and bottom-released particles; yet, these differences are so substantial in the simulation results that it seems that most particles did not leave the surface or bottom model layer. Were the particles tied to the surface or bottom layer (i.e. vertical velocities set to zero)? It does not seem so, because at some point in the manuscript up- and down-welling are addressed. Was the effect of turbulent mixing omitted? Why? If so, I'm not sure what the presented results mean/represent. If the effects of (vertical) turbulent mixing were included, checks need to be made to see if this was done correctly. This needs to be resolved and/or absolutely clear, otherwise this work cannot be published.

Missing information (main points)

-It is not clearly explained a priory (section 2.4) why each experiment was carried out, and with which objective (what do you expect to learn and how will these experiments provide that knowledge)?

-There is no validation of the temperature and salinity fields. The salinity fields correspond to what I would expect, but the temperature fields don't.

-l. 54-62. Also discuss seasonal stratification and subsurface jets (eg. Hill ea 2008).

-Different, and offline model for backtracking: a forward run of this model for the control run should be compared with CR to identify/quantify differences in results resulting from the model/method differences.

-Markings in the figures (plusses, minuses, stars, arrows): It is not clear what the criteria were to place these where they were put (often other locations seem equally justifiable), or for which areas they hold. Please remove and find a different/better/more quantitative way to quantify/visualise/discuss this.

-There is repeated mentioning of 'westerlies' as an explanation in the discussion/conclusions, but no detail about the wind forcing is presented (e.g. to show that this was the dominant wind direction during the simulations), nor short-term simulations (e.g. to show what happens when the wind is from the west).

-Section 3.4. Why were 80m intervals chosen? Why at these depths? Why allow gaps in the vertical in this analysis? What happens in the gaps? It would help if the initial positions were shown?

-Figure 2: for which depths are these data presented? Surface, bottom, depth-averaged?

-Monthly runs were done to depict DT. How dependent is the result on this monthly interval? For instance, what happens to the visualised results if 2 weeks or 2 months

are used as interval?

Structure

-Section 2.2 contains results, please separate methods and results.

-Section 2.3 contains (many) results, please separate methods and results.

-Results and discussion: there does not seem to be much system in the order in which the various release experiments (as in Table 2) are presented/discussed, with quite a few jumps between experiments. There also does not seem to be much balance in the amount of attention given to these various experiments. This makes it difficult for the reader to keep track of the narrative. This should be tidied up; one way of doing that would be to split the section into two separate Results and Discussion sections. Linking back to the objectives of each experiment (see also Missing information above) will also help here.

Density Trend

I don't think that this term describes the quantity properly. For instance, 'trend' typically indicates a change in time, which is not the case here. Please find a better descriptor. I would suggest 'Normalised Cumulative (Particle) Density'?

Detailed comments

l. 29. bracket missing

l. 29. '..achieved with substantial contributions from Eulerian numerical...'

l. 31-33. I don't understand why this remark is made, see l. 64-69.

l. 41. or around S/W Ireland.

l. 44. Baltic, subsequently those

l. 96. value horizontal eddy viscosity: please check value. It seems odd that it is negative, and to the power of 10?

l. 98. Drag coefficient: units?

l. 99. output: and vertical velocity?

l. 105. spinup period: is this enough? The North Sea has a residence time of several years.

l. 109-110. Really? Please check: I don't think hourly data are available.

l. 112. operational FOAM-AMM7.

l. 155. The residual velocities and the velocity amplitudes... Please use the term 'residual velocities' throughout for this element (I will not indicate all occurrances).

l. 158. low residual velocities

l. 159. surely not only in the English Channel?

l. 160. It is not clear to me how this is defined/quantified?

l. 163. 'sea level oscillations': these are not presented: velocity oscillations?

l. 174. Why winter? How is 'winter' defined?

l. 177. The East Anglia plume is not defined by temperature, but by turbidity (and lies somewhat south of the location suggested here).

l. 178. Frisian front: is a summer feature separating temperature-stratified from well-mixed conditions.

l. 182. 'a number of mesoscale features': it is not clear to me what is meant here.

l. 183. '...compare with...': it is not clear to me what should be compared.

l. 184. Fig S2a: why is this fig in the supplementary material?

l. 188. NTE: why does the narrative jump to this here? What about stratification?

l. 189. 'much of what is known': please specify.

l. 192-205: please better specify/explain variables.

l. 206. Why 25 h average (I can guess, but not everyone might). On which day? Why this day? Why one day?

l. 212. difference: which was subtacted from which?

l. 213. 'despite': replace by 'In addition to'

l. 212-214: what do positive/negative values mean?

l. 216. 'significant importance': what does it do to them?

l. 217. this is not presented?

l. 218. I don't understand this sentence/reasoning.

l. 229. This is a conclusion. Also: how can you tell, as the experiments were not set up in the same way?

l. 236. ARIANE user manual: please provide reference.

l. 237-239. Then how, exactly, does the horizontal particle diffusion work? Is the 7 km grid really sufficient to explicitly resolve all horizontal turbulent diffusion processes as eddies?

l. 245: stripe: strip. Along the shelf edge? A figure may help here.

l. 248. Why use a different model?

l. 250. Why at a constant depth, i.e. different (?) from the forward experiments? How can you then compare? Why at 1 m, and not at the surface as in the other experiments?

l. 285. 6 months? Why?

l. 259. only january: why?

l. 260. NTE also July: why?

Table 2. Please complete with release time. Backtracking: 1 m, not surface.

l. 305. Why 12 h? The tidal period is (roughly) 12.5 h, so the difference between start and end point of the depicted loops are not the residual (or net transport), but still contain a tidal contribution.

l. 318. Refer to fig 4 after 'different'.

Figure 4. After introducing DT, it is not clear to me why the plots of particle positions were included? If it is to point out that DT is a better way to visualise, one simple comparison figure should suffice.

l. 324. It is not clear to me what the authors aim to point out here?

l. 331. breaking internal waves: the hydrostatic NEMO model cannot represent these.

l. 333. I don't see the causal relationship here?

l. 342. flown: been transported? They don't have wings...

l. 355. ambiguous: what is meant with this? If you mean that the accumulation patterns have high spatial variability (or something like that), then say that? Please also change other occurrences?

l. 366. off-shore

l. 372. reduced: smaller than for surface particles

l. 382. I'm not sure what exactly you're indicating here.

l. 397. Irish Sea. Why mention this specifically: there are other places, too.

l. 412. 'possibly indicating': can you quantify the scales to make this a firm statement?

l. 418. the front: which front? Also I'm not sure if there is a front in winter?

l. 421. suggests: how?

l. 427. variability: of what?

l. 429. FEW: FWE?

l. 435. 'smoothing': please be consistent (with the abbreviation), and use 'filtering' throughout.

l. 440. 'that disappears': one can't see this in difference plots?

l. 442. this contradicts the previous sentence.

l. 445. Remove of substantiate.

l. 453. 'differences': please specify.

l. 469. 'side of particle supply': what exactly do you mean?

l. 470. 'the particle supply is hampered by the front': what exactly do you mean?

l. 468-470. So backtracking experiments do not produce realistic results, as interactions with frontal dynamics are non-reversible?

l. 483-487. Please demonstrate this by providing wind data.

l. 491. How could this work? Most fronts are absent in January.

l. 512. I don't understand this sentence.

l. 528. So what is causing the up/downwelling there, then?

l. 548. So what does this experiment add?

l. 578. 'thalweg': This is German, please find English equivalent. Also occurs elsewere.

l. 587. 'floating marine debris': only floating?

l. 589 etc.: Please provide links/references to data sources.

Figure captions: please put graph labels before the descriptors, not after.

Figure 3, caption: what are the isobaths in a) and b)?

Figure 5, 7 caption: southern bight, not German Bight, please check throughout.

Figure 5: 'annual mean' is depicted, not 'monthly average'?

Figure 7: distance: along transect?

Figure 8, caption: I'm not sure what's meant with the last sentence.

Figure 9. It is not clear to me why a portion of the particles is purple? Surely they have all potentially changed depth?

---

## Referee Comment (RC2) · Anonymous Referee #2 · 24 Jan 2020

Summary

The authors have performed a set of Lagrangian particle tracking experiments to study the water circulation on the European Northwest Shelf (ENWS). Several scenarios were simulated, with particles (passive tracers, or water masses) released at surface and seafloor, and simulated forwards for up to 1 year, plus one case with backwards simulations. A property called "density trend" is defined to aid the analysis of the spatial accumulation of particles.

General comments

As the authors themselves point out, several modeling studies have looked at the ENWS, but not so many studies have applied Lagrangian methods, at least not for

the whole area. The simulated scenarios are sensible, and the discussion contains several interesting comments and findings, though nothing groundbreaking. The main weakness of the paper is that the discussion would need a more clear structure, and be better linked to well defined motivation/objectives. But after improving the structure (i.e. major revision) and some details as discussed below, I would find this manuscript suitable for publication.

Specific comments

Line 29: Missing end parenthesis.

Lines 40-50 discusses typical current patterns. It would be helpful with a figure with arrows to better follow this description.

Line 50: Could ref to Fig2c for the comment about low salinity along coast.

Lines 50-52: This major hypothesis should be reflected also in abstract.

Lines 60-62: Sentence is a bit hard to read.

Line 75: Should mention here that vertical mixing is also not considered. This is an important point, that should also be discussed/justified.

Line 89: It is not clear whether the area of Fig 1 is identical to the AMM7 area, or if this is a subset?

Line 90: AMM7 is called a model, but perhaps "model setup" is more precise?

Line 93: Here the term "tracer" is used. It should be made clear whether tracer and particles are the same thing in this study.

Line 95: Please provide a reference or justification for the choice of eddy diffusivity. It should be commented that this is constant throughout the area (which is not true in reality).

Line 96: Eddy viscosity should be a positive number.

[Figure]

Section 2.2: More information should be given about the drifter type/characteristics/name, as near-surface drifters are affected by a varying degree of Stokes drift and wind drag, see e.g. Röhrs, J., K. H. Christensen, L. R. Hole, G. Broström, M. Drivdal, and S. Sundby (2012), Observation-based evaluation of surface wave effects on currents and trajectory forecasts, Ocean Dyn., 62, 1519–1533

Thus, a missing contribution from Stokes drift can possibly explain why the model currents are too slow in the comparison. Alternatively, SVP drifters (15m depth) from the Global Drifter Program could be used to validate the model current, so that Stokes drift would not be an issue. Also a plot of the complete drifter trajectories should be shown, to justify whether they cover a substantial part of the area, or just locally to their deployment location.

Line 148/Table1: The number of comparison points should be provided.

Section 2.3. This discussion is a bit messy, and does also belong in the results section, rather than under "material and methods".

Line 184: It could be made clear (the first time) that Figure S2a refers to figure 2a in the supplements.

Line 207: could be commented that the Molinari and Kirway study is for the Caribbean during summer, thus quite different conditions.

Line 240: It should be commented (and discussed) that vertical mixing is not included.

Line 242: Should also be mentioned here that particles are released over the whole domain.

Line 244-246: The seeding locations of CR-V should also be shown on a figure

Line 248: It should be mentioned explicitly that a separate offline trajectory model has to be used for the backwards simulations, as this is not possible to do with online simulations. However, a forward simulation with this offline model should also be done

to benchmark it against the online forward simulations.

Lines 274-279: What would be the difference between "density trend" and "residence time"?

Line 278: "motionless situation" is a bit unclear, please rewrite sentence.

Line 302-304: Please clarify what is meant here.

Line 461: extra space after "Channel"

Section 3 is a bit lengthy, and hard to read due to jumping back and forth between the experiments and referring to many figures. Making it a bit more compact and structured would help.

Figures

There are a lot of composite figures/maps of the area of interest. These are quite small and hard to read when printed on A4 paper. Could whitespace be reduced somehow?

In the figure captions, the letters a), b)... should rather be placed before the explanation, and not after

Figure 2: CR and NTE should be written explicitly as "control run" and "no tides experiment", so that the figure can be read and understood also before reading the main text. Same for other figures. Line 847: und -> and

Figure 3: a bit much spaghetti here, perhaps use even fewer than every 5th trajectory?

Figure 4: Caption is quite hard to read. The '+' and '-' symbols are presumably placed "by hand"? This is generally ok, but they are quite many, and sometimes slightly displaced, perhaps to avoid overlap? So in practice I don't think these symbols work very well here. Could the point be visualized by another, more objective measure?

Figure 5: Title of lower figure is "monthly average", but I guess it should be "yearly average", or "average of months"

**References**

Please update this reference, where you refer to a discussion paper: Dagestad, K.-F., Röhrs, J., Breivik, Ø., and Ådlandsvik, B.: OpenDrift v1.0: a generic framework for trajectory modelling, Geosci. Model Dev., 11, 1405–1420, https://doi.org/10.5194/gmd-11-1405-2018, 2018.

---

## Author Comment (AC1) · 11 Mar 2020

Review of 'Circulation of the European Northwest Shelf: A Lagrangian perspective' by Marcel Ricker and Emil Stanev.

*We thank reviewer #1 for the detailed and extensive review of our paper. We provide point-by-point answers in the attached pdf.*

Summary

The manuscript describes a series of Lagrangian particle tracking experiments carried out to characterise the water circulation and the accumulation of hypothetical particles on the northwest European continental shelf. Particles were released at the surface and the bottom at the start of monthly runs for six different scenarios, mostly uniformly distributed over the area. Some scenarios were carried out over one year, others only for January. A property called 'density trend' is defined to analyse the resulting particle movements. The analysis consists mainly of differences of this 'density trend' between the model scenarios.

General comments

This is an interesting manuscript, with powerful visualisations, supported to a large extent by the newly introduced 'density trend'. However, after reading it, I am left mostly confused, because important information is not supplied (or has escaped my attention), and choices are not motivated (while some of these choices are bound to affect the results). Moreover, the results and discussion section is rather long-winded, depends on visual comparison between figures (which will end up on different pages), and seems to jump unpredictably between figures in quite a few places. It is not clear to me how the particles (are allowed to) move in the vertical (vertical mixing seems to be absent or under-represented). For a number of figures, it is not clear if surface, bottom or e.g. depth-averaged values are presented. Also, I have doubts about the January temperature fields presented/used: the temperatures seem too high, and I am surprised by the magnitude of the spatial gradients in the North Sea. Validation of these is absent. Overall, I would recommend major revisions. I will provide more detailed comments below.

Authors: In the following we provide the missing information and explain the questionable choices. We hope that the changes we introduce contribute to a better understanding of the manuscript. We rearranged the order of the manuscript and of the experiments. A detailed comment on the vertical particle movement is given in the following paragraph. The model has been re-calibrated and a comparison of surface temperatures and satellite data is shown in the supplementary material. Hence, ALL figures have been updated. As suggested, the quantity "density trend (DT)" has been renamed to "normalised cumulative particle density (NCPD)".

Vertical particle motion

Under well-mixed conditions, as may be expected in most of the North Sea in winter, I would expect neutrally buoyant particles (as simulated here?) to be mixed quickly in the vertical by turbulent mixing processes. That means that there should be little difference in the overall dispersal of surface and bottom-released particles; yet, these differences are so substantial in the simulation results that it seems that most particles did not leave the surface or bottom model layer. Were the particles tied to the surface or bottom layer (i.e. vertical velocities set to zero)? It does not seem so, because at some point in the manuscript up- and down-welling are addressed. Was the effect of turbulent mixing omitted? Why? If so, I'm not sure what the presented results mean/represent. If the effects of (vertical) turbulent mixing were included, checks need to be made to see if this was done correctly. This needs to be resolved and/or absolutely clear, otherwise this work cannot be published.

Authors: We made clear in the revised manuscript that the used Lagrangian techniques aim at giving a new view on velocity field in the North Sea. In other words, the paper is about velocity, not so much about turbulence. We do not analyse the propagation and mixing of particles. In our setup, particles released in NEMO are always advected in 3-D by (u,v,w). That is, the particles are neutrally buoyant (added in line 163-164) and can be interpreted as following the pathways of water parcels (Blanke and Raynaud, 1997). Because we study the properties of the velocity field, additional horizontal and vertical turbulent mixing is not introduced for particle tracking. As a consequence, the presented analyses are analyses of velocity properties and not of the effects of mixing (added in line 83-84 and 163-164).

Nevertheless, in terms of T/S, the water column is well mixed in January, thus the model physics can be treated as correct. The specific properties of the velocity field explains the difference of NCPD at the surface and bottom.

Implementations of turbulent mixing in Lagrangian tracking is mostly done by random walk schemes. The effect of horizontal diffusion is shown in Fig. R.1.1. We want to emphasise that the implementation of horizontal (van Sebille et al., 2018) and, in particular, vertical diffusion (van Sebille et al., 2020) in particle tracking are ongoing scientific subjects.

van Sebille, E., Aliani, S., Law, K. L., Maximenko, N., Alsina, J., Bagaev, A., et al. (2020). The physical oceanography of the transport of floating marine debris. Environmental Research Letters. https://doi.org/10.1088/1748-9326/ab6d7d

[Figure]

**Fig. R1.1**. Surface January 2015 NCPD without (left) and with (middle) additional horizontal diffusion in particle advection obtained from offline simulations. The right panel shows the difference without minus with diffusion.

Missing information (main points)

-It is not clearly explained a priory (section 2.4) why each experiment was carried out, and with which objective (what do you expect to learn and how will these experiments provide that knowledge)?

Authors: More details about the experiments and their objectives are added in the text (line 188-196).

-There is no validation of the temperature and salinity fields. The salinity fields correspond to what I would expect, but the temperature fields don't.

Authors: The model has been further tuned to better represent the thermohaline fields. A figure has been added in the supplementary material showing the RMSE of satellite and model data in January 2015 (Fig. S3; line 318-320).

-l. 54-62. Also discuss seasonal stratification and subsurface jets (eg. Hill ea 2008).

Authors: Seasonal stratification was already mentioned. Subsurface jets are referenced by citing Hill et al. (2008) (line 57-61).

-Different, and offline model for backtracking: a forward run of this model for the control run should be compared with CR to identify/quantify differences in results resulting from the model/method differences.

Authors: Such comparison has been made during the preparation of the manuscript and is mentioned in the text in line 151-153. The comparison in terms of NCPD using the results from

an online run without vertical advection and the same setup in OpenDrift for January 2015 is shown in Fig. R1.2 (NCPD online minus offline). The differences are rather minor.

[Figure]

**Fig. R1.2**. Surface January 2015 NCPD online 2-D (left), offline (middle) and the difference online minus offline (right).

-Markings in the figures (plusses, minuses, stars, arrows): It is not clear what the criteria were to place these where they were put (often other locations seem equally justifiable), or for which areas they hold. Please remove and find a different/better/more quantitative way to quantify/visualise/discuss this.

Authors: For Fig. 4, NCPD can be interpreted as a quantitative measure for particle accumulation. Thus, we decided to avoid any of these markers and we emphasise, that we describe examples of pronounced features (line 374-375 and 565).

-There is repeated mentioning of 'westerlies' as an explanation in the discussion/ conclusions, but no detail about the wind forcing is presented (e.g. to show that this was the dominant wind direction during the simulations), nor short-term simulations (e.g. to show what happens when the wind is from the west).

Authors: During the preparation of the manuscript, analyses of the wind have been made. To remove ambiguity also for other readers, a wind rose of these analyses has been added in the supplementary material (Fig. S1; line 252-253). The wind contains (almost) always a western component.

-Section 3.4. Why were 80m intervals chosen? Why at these depths? Why allow gaps in the vertical in this analysis? What happens in the gaps? It would help if the initial positions were shown?

Authors: We are sorry about the confusion. There are no gaps. Particles are seeded every 20 m, i.e. at 1 m, 20 m, 40 m, … down to the bottom. Thus, the results in Fig. 9a are for the 1 m, 20 m, …, and 80 m seeding. The next seeding levels are 100 m, 120 m, …, and 180 m (Fig. 9b). So the gap in the labels is misleading and it should read 1-100 m, 100-200 m, 200-300 m

and 900-1,000 m. An example of initial positions have been added in Fig. 9e. In the horizontal, the particles are seeded uniformly within the coloured area (1 per model grid).

-Figure 2: for which depths are these data presented? Surface, bottom, depthaveraged?
Authors: As given in the caption ("simulated surface properties"), the data is always shown for the surface (1 m) for the comparisons with surface NCPD.

-Monthly runs were done to depict DT. How dependent is the result on this monthly interval? For instance, what happens to the visualised results if 2 weeks or 2 months are used as interval?
Authors: 2 weeks and 2 months as integration time would lead to similar results (see Fig. R1.3; compare with Fig. 5a). In 2 weeks the results are noisier and accumulation areas less pronounced; 2 months already average out small-scale features. If the integration time is too long, vast areas show very low NCPD which results from "empty" grids that are no longer supplied by particles. For these grids (white grids in NCPD plots), the statistical relevance cannot be ensured.

[Figure]

**Fig. R1.3**. Surface NCPD for the first 15 (left) and 60 days (right) in 2015.

Structure
-Section 2.2 contains results, please separate methods and results.
-Section 2.3 contains (many) results, please separate methods and results.
Authors: The results of Sect. 2.2 and 2.3 have been moved into the results (note the rearranged order of the manuscript).

-Results and discussion: there does not seem to be much system in the order in which the various release experiments (as in Table 2) are presented/discussed, with quite a few jumps between experiments. There also does not seem to be much balance in the amount of attention given to these various experiments. This makes it difficult for the reader to keep track of the narrative. This should be tidied up; one way of doing that would be to split the section into two

separate Results and Discussion sections. Linking back to the objectives of each experiment (see also Missing information above) will also help here.

Authors: The structure of the manuscript has been revised. We tried to split the sections into equally long sections, whereas the balance between the experiments was not considerably changed, because some experiments require more attention (e.g. CR) and some less (e.g. CR-B). We also reordered the experiments according to their appearance in the text; same for the supplementary figures. The jumping between experiments and figures results from a manuscript structure which is based on certain physical topics. Therefore, it is inevitable to refer only to one experiment. The figure references are thought to help to orientate while jumping back and forth. The objectives of the experiments are commented above.

Density Trend

I don't think that this term describes the quantity properly. For instance, 'trend' typically indicates a change in time, which is not the case here. Please find a better descriptor. I would suggest 'Normalised Cumulative (Particle) Density'?

Authors: We appreciate this suggestion and changed "density trend (DT)" to "normalised cumulative particle density (NCPD)".

Detailed comments

l. 29. bracket missing

Authors: Added.

l. 29. '..achieved with substantial contributions from Eulerian numerical...'

Authors: Done as suggested.

l. 31-33. I don't understand why this remark is made, see l. 64-69.

Authors: Changed (see line 33).

l. 41. or around S/W Ireland.

Authors: Done as suggested.

l. 44. Baltic, subsequently those

Authors: Done as suggested.

l. 96. value horizontal eddy viscosity: please check value. It seems odd that it is negative, and to the power of 10?

Authors: Please, keep in mind that we talk about bi-harmonic mixing. For comparison, in O'Dea et al. (2017) the same viscosity was chosen.

l. 98. Drag coefficient: units?

Authors: The drag coefficient is non-dimensional, but the roughness length is not, which has been added together with the allowed drag coefficient range (line 107-108).

l. 99. output: and vertical velocity?

Authors: We made some analyses of the vertical velocities, but they did not revealed anything new and are thus not shown.

l. 105. spinup period: is this enough? The North Sea has a residence time of several years.

Authors: Please, see the below Fig. R1.4.

[Figure]

**Fig. R1.4**. Monthly EKE at 100 m averaged over the whole domain for 2014 and 2015.

l. 109-110. Really? Please check: I don't think hourly data are available.

Authors: It seems that for AMM7 hourly instantaneous data is no longer available on the Copernicus website (only daily means). Thanks for the hint. The text has been updated accordingly (line 121).

l. 112. operational FOAM-AMM7.

Authors: Done as suggested.

l. 155. The residual velocities and the velocity amplitudes... Please use the term 'residual velocities' throughout for this element (I will not indicate all occurrances).

Authors: Done as suggested.

l. 158. low residual velocities

Authors: Done as suggested.

l. 159. surely not only in the English Channel?

Authors: Extended (line 246).

l. 160. It is not clear to me how this is defined/quantified?

Authors: We found, as described in the text, that the model dynamics significantly change along this isobath.

l. 163. 'sea level oscillations': these are not presented: velocity oscillations?

Authors: "Sea level oscillations" is correct and relates to the previous sentence in the manuscript. SSH patterns were analysed but did not show new aspects and do not support the following analyses; thus they were discarded for the final set of figures.

l. 174. Why winter? How is 'winter' defined?

Authors: "winter" has been replaced by January and "summer" by July.

l. 177. The East Anglia plume is not defined by temperature, but by turbidity (and lies somewhat south of the location suggested here).

Authors: This is correct. The formulation was not appropriate and has been changed (line 268-271).

l. 178. Frisian front: is a summer feature separating temperature-stratified from wellmixed conditions.

Authors: The Frisian Front has been moved to July (line 274-276).

l. 182. 'a number of mesoscale features': it is not clear to me what is meant here.

Authors: This term relates to the patterns of the temperature gradients (Fig. 2f) in the deep ocean. They coincide with the patterns of velocity magnitudes (Fig. 2a) and have sizes of tens of kilometres.

l. 183. '...compare with...': it is not clear to me what should be compared.

Authors: Done as suggested.

l. 184. Fig S2a: why is this fig in the supplementary material?

Authors: For a clearer structure, the figures in the main manuscript only show January. To give an impression of the situation in July, too, we put all of these results on one page in the supplementary material.

l. 188. NTE: why does the narrative jump to this here? What about stratification?

Authors: The NTE is mentioned to proof the influence of tides. Stratification is also different in the NTE and CR, that is, the velocity field is also affected by stratification.

l. 189. 'much of what is known': please specify.

Authors: Done as suggested (line 280).

l. 192-205: please better specify/explain variables.

Authors: Done as suggested (line 297-298).

l. 206. Why 25 h average (I can guess, but not everyone might). On which day? Why this day? Why one day?

Authors: The explanation for using a 24.84 h-period (changed from 25 h) is added to the text (line 298-300). The day (15.01.2015) is given in the figure caption but is also added in the text.

l. 212. difference: which was subtacted from which?

Authors: The text reads "difference … between the CR and NTE", i.e. CR minus NTE. To make it clear, the caption of Fig. 2 and the text (line 306) have been improved.

l. 213. 'despite': replace by 'In addition to'

Authors: Done as suggested.

l. 212-214: what do positive/negative values mean?

Authors: Positive/negative values mean reduction/increase of deformation by tides.

l. 216. 'significant importance': what does it do to them?

Authors: The presence of deformation indicates the possible contribution of shear to horizontal mixing, e.g. Sanderson and Okubo (1986 and 1987).

Sanderson, B. G., & Okubo, A. (1986). An analytical calculation of two-dimensional dispersion. *Journal of the Oceanographical Society of Japan*, *42*(2), 139–153. https://doi.org/10.1007/BF02109101

Sanderson, B. G., & Okubo, A. (1987). Comments on the "shear effect" and diffusion in the Lagrangian framework. *Journal of the Oceanographical Society of Japan*, *43*(3), 183–196. https://doi.org/10.1007/BF02109218

l. 217. this is not presented?

Authors: Not for January, because the difference is shown instead. For July it is shown (Fig. S2d).

l. 218. I don't understand this sentence/reasoning.

Authors: The inner Bay of Biscay is mainly dominated by mesoscale dynamics, e.g. eddies. If the deformation difference plot (Fig. 2h) shows patterns with approximately the size of eddies, it can be concluded that they cause them.

l. 229. This is a conclusion. Also: how can you tell, as the experiments were not set up in the same way?

Authors: The intercomparison was done with the same setup (2-D online advection) and showed only very small differences (see previous answer). To prevent doubts about the similarity of online and offline simulations, a short explanation about the used setups is given in the text (line 152).

l. 236. ARIANE user manual: please provide reference.

Authors: The website has been added (line 158-159).

l. 237-239. Then how, exactly, does the horizontal particle diffusion work? Is the 7 km grid really sufficient to explicitly resolve all horizontal turbulent diffusion processes as eddies?

Authors: As already said, there is no consensus about adding extra lateral diffusion to particle tracking (van Sebille et al., 2018). Furthermore, our tests showed that extra diffusion will lead to slightly noisier NCPD fields but will not change the patterns substantially (this was tested in previous experiments; see also Fig. R1.1). Concerning what the model can resolve, we give examples from Stanev and Ricker (2020) where this issue is explained in detail (line 98-101).

Stanev, E. V., & Ricker, M. (2020). Interactions between barotropic tides and mesoscale processes in deep ocean and shelf regions. Accepted in *Ocean Dynamics.*

l. 245: stripe: strip. Along the shelf edge? A figure may help here.

Authors: We use "stripe" throughout the text both having the same meaning. Fig. 9 has been referenced (line 176-177).

l. 248. Why use a different model?

Authors: Backtracking is not possible online. This has been mentioned in the text (line 151).

l. 250. Why at a constant depth, i.e. different (?) from the forward experiments? How can you then compare? Why at 1 m, and not at the surface as in the other experiments?

Authors: The backtracking experiment is rather a complement of the forward experiments and is not thought to be compared in detail with the forward experiments. Despite that, in the online simulation, the seeding is also at 1 m.

l. 258. 6 months? Why?

Authors: This was done for Fig. 4. After 6 months the resulting patterns can be seen nicely and contributes an impression of particle accumulation/dispersal for longer time scales than one month.

l. 259. only january: why?

Authors: The focus of the paper is on monthly time scales. As an example January was chosen. Due to the importance of stratification in summer, July was chosen as a second example (see comment below) and put into the supplementary material. Furthermore, changes in tidal and wind forcing are applied on 01 January 2015. Impacts of the changes in forcing on the dynamics can only directly be seen during the transition period. Analyses of later months would rather describe the new steady state.

l. 260. NTE also July: why?

Authors: To show a representative summer pattern.

Table 2. Please complete with release time. Backtracking: 1 m, not surface.

Authors: The caption has been improved. Note the changed order of Table 1 and 2.

l. 305. Why 12 h? The tidal period is (roughly) 12.5 h, so the difference between start and end point of the depicted loops are not the residual (or net transport), but still contain a tidal contribution.

Authors: This is correct. Thus, a 12.42 h period was chosen to show a full $M_2$ cycle. Further, it is mentioned that the "The difference between the start and end positions on the circular loops gives an estimate of the net transport, …" (line 357-358).

l. 318. Refer to fig 4 after 'different'.

Authors: Done as suggested.

Figure 4. After introducing DT, it is not clear to me why the plots of particle positions were included? If it is to point out that DT is a better way to visualise, one simple comparison figure should suffice.

Authors: As correctly mentioned, the advantage of NCPD is the better way to visualise particle accumulation. Nevertheless, Fig. 4 also shows important details, especially the plots after 6 months. Particle positions also contribute to a better understanding of NCPD. This is mentioned in the paragraph starting at line 410.

l. 324. It is not clear to me what the authors aim to point out here?

Authors: Now, the sentence refers to Sect. 4.3, where this point is discussed in detail.

l. 331. breaking internal waves: the hydrostatic NEMO model cannot represent these.

Authors: "breaking" removed.

l. 333. I don't see the causal relationship here?

Authors: There are several dynamic processes prevailing at the continental slope. Most of them induce a net transport which in turn affect the particle dynamics. The text has been improved (line 382).

l. 342. flown: been transported? They don't have wings...

Authors: Done as suggested.

l. 355. ambiguous: what is meant with this? If you mean that the accumulation patterns have high spatial variability (or something like that), then say that? Please also change other occurrences?

Authors: Done as suggested.

l. 366. off-shore

Authors: Done as suggested.

l. 372. reduced: smaller than for surface particles

Authors: Done as suggested.

l. 382. I'm not sure what exactly you're indicating here.

Authors: Extended (line 436-437).

l. 397. Irish Sea. Why mention this specifically: there are other places, too.

Authors: Yes, they are listed at the end of the paragraph and in the following ones.

l. 412. 'possibly indicating': can you quantify the scales to make this a firm statement?

Authors: Done as suggested (line 464).

l. 418. the front: which front? Also I'm not sure if there is a front in winter?

Authors: The flanks of the warm water plume from the English Channel were meant. The text has been improved accordingly (line 470).

l. 421. suggests: how?

Authors: Extended (line 473-474).

l. 427. variability: of what?

Authors: Extended (line 478).

l. 429. FEW: FWE?

Authors: Changed.

l. 435. 'smoothing': please be consistent (with the abbreviation), and use 'filtering' throughout.

Authors: Done as suggested.

l. 440. 'that disappears': one can't see this in difference plots?

Authors: This is true and has been replaced (line 491).

l. 442. this contradicts the previous sentence.

Authors: We don't see a contradiction. Vast shelf areas are affected in terms of bottom particle accumulation but the most affected ones are in the northern shelf and Norwegian Trench/Skagerrak. The text has been modified to clarify that (line 493).

l. 445. Remove of substantiate.

Authors: These differences are important but can also be seen by the comparison of Fig. 6e and 6g and Fig. 6f and 6h, respectively. Thus, these figures have not been shown. The hint to compare the respective figures has been added (line 497).

l. 453. 'differences': please specify.

Authors: Explained (line 506).

l. 469. 'side of particle supply': what exactly do you mean?

Authors: An explanation was is given in the previous sentence but has been extended for clarification (line 522-524).

l. 470. 'the particle supply is hampered by the front': what exactly do you mean?

Authors: Sentence added (line 522-524).

l. 468-470. So backtracking experiments do not produce realistic results, as interactions with frontal dynamics are non-reversible?

Authors: For areas with strong fronts and areas downstream this is true. This point has been added to the text (line 524-525).

l. 483-487. Please demonstrate this by providing wind data.

Authors: Please see Fig. S1.

l. 491. How could this work? Most fronts are absent in January.

Authors: Thanks, this was wrong (sentence removed).

l. 512. I don't understand this sentence.

Authors: The sentence means, that a surface NCPD <1 (=particle dispersal) and a bottom NCPD >1 (=particle accumulation) should be related to an upward movement of water similar to positive and negative divergence.

l. 528. So what is causing the up/downwelling there, then?

Authors: This is explained in the following sentence (line 584).

l. 548. So what does this experiment add?

Authors: A sentence in the conclusions has been added (line 615-617).

l. 578. 'thalweg': This is German, please find English equivalent. Also occurs elsewere.

Authors: Although "Talweg" is the German word, "thalweg" is the English translation and is commonly used in this sense.

l. 587. 'floating marine debris': only floating?

Authors: Actually it has to be "neutrally buoyant floating marine debris which is released at the sea surface and sea floor". These characteristics should be clear from the manuscript and thus it is abbreviated as "floating marine debris".

l. 589 etc.: Please provide links/references to data sources.

Authors: The links and references for data and models have been shifted from the text to this section.

Figure captions: please put graph labels before the descriptors, not after.

Authors: Done as suggested.

Figure 3, caption: what are the isobaths in a) and b)?

Authors: The black isobath (200 m) is mentioned in Fig. 1 and is the same in all figures. For Fig. 3c and d the isobaths are mentioned in the caption of Fig. 3.

Figure 5, 7 caption: southern bight, not German Bight, please check throughout.

Authors: Thanks for the hint. Done as suggested.

Figure 5: 'annual mean' is depicted, not 'monthly average'?

Authors: Yes, changed accordingly.

Figure 7: distance: along transect?

Authors: Yes, for clarity, a sentence has been added in the caption.

Figure 8, caption: I'm not sure what's meant with the last sentence.

Authors: It is the caption for (b) (=CR-B results). Should be clear after changing the position of the graph levels.

Figure 9. It is not clear to me why a portion of the particles is purple? Surely they have all potentially changed depth?

Authors: All particles of this experiment are purple. Purple was chosen to differentiate them from colours appearing in the colour bar. In each plot, the purple dots show the particles used to calculate the depth change of the respective figure (blue-red-coloured grids). The depth change is shown at the initial particle positions.

---

## Author Comment (AC2) · 11 Mar 2020

We thank reviewer #2 for the constructive review of our paper. We provide point-by-point answers in the attached pdf.

Summary

The authors have performed a set of Lagrangian particle tracking experiments to study the water circulation on the European Northwest Shelf (ENWS). Several scenarios were simulated, with particles (passive tracers, or water masses) released at surface and seafloor, and simulated forwards for up to 1 year, plus one case with backwards simulations. A property called "density trend" is defined to aid the analysis of the spatial accumulation of particles.

General comments

As the authors themselves point out, several modeling studies have looked at the ENWS, but not so many studies have applied Lagrangian methods, at least not for the whole area. The simulated scenarios are sensible, and the discussion contains several interesting comments and findings, though nothing groundbreaking. The main weakness of the paper is that the discussion would need a more clear structure, and be better linked to well defined motivation/objectives. But after improving the structure (i.e. major revision) and some details as discussed below, I would find this manuscript suitable for publication.

Authors: Thank you. We improved the structure (see also a comment below) of the paper and clarified the objectives of the respective experiments (line 188-196). The minor comments have been implemented as given below. Note that the quantity "density trend (DT)" has been renamed to "normalised cumulative particle density (NCPD)".

Specific comments

Line 29: Missing end parenthesis.

Authors: Added.

Lines 40-50 discusses typical current patterns. It would be helpful with a figure with arrows to better follow this description.

Authors: We added schematically grey arrows in Fig. 1 to indicate the general shelf sea circulation (line 42-43). To complement this, we added "Howarth (2001)" (line 52-53) as a reference for North Sea circulation.

Line 50: Could ref to Fig2c for the comment about low salinity along coast.

Authors: To improve the structure, we want to prevent mixing up the order of the figures appearances. However, Fig. 2d is mentioned in Sect. 2.3 with respect to the low salinity along the coasts.

Lines 50-52: This major hypothesis should be reflected also in abstract.

Authors: Done as suggested (line 24).

Lines 60-62: Sentence is a bit hard to read.

Authors: Improved (line 65).

Line 75: Should mention here that vertical mixing is also not considered. This is an important point, that should also be discussed/justified.

Authors: We made clear in the revised manuscript that the used Lagrangian techniques aim at giving a new view on velocity field in the North Sea. In other words, the paper is about velocity, not so much about turbulence. We do not analyse the propagation and mixing of particles. In our setup, particles released in NEMO are always advected in 3-D by (u,v,w). That is, the particles are neutrally buoyant (added in line 163-164) and can be interpreted as following the pathways of water parcels (Blanke and Raynaud, 1997). Because we study the properties of the velocity field, additional horizontal and vertical turbulent mixing is not introduced for particle tracking. As a consequence, the presented analyses are analyses of velocity properties and not of the effects of mixing (added in line 83-84 and 163-164).

Nevertheless, in terms of T/S, the water column is well mixed in January, thus the model physics can be treated as correct. The specific properties of the velocity field explains the difference of NCPD at the surface and bottom.

Implementations of turbulent mixing in Lagrangian tracking is mostly done by random walk schemes. The effect of horizontal diffusion is shown in Fig. R.2.1. We want to emphasise that the implementation of horizontal (van Sebille et al., 2018) and, in particular, vertical diffusion (van Sebille et al., 2020) in particle tracking are ongoing scientific subjects.

van Sebille, E., Aliani, S., Law, K. L., Maximenko, N., Alsina, J., Bagaev, A., et al. (2020). The physical oceanography of the transport of floating marine debris. Environmental Research Letters. https://doi.org/10.1088/1748-9326/ab6d7d

[Figure]

**Fig. R2.1**. Surface January 2015 NCPD without (left) and with (middle) additional horizontal diffusion in particle advection obtained from offline simulations. The right panel shows the difference without minus with diffusion.

Line 89: It is not clear whether the area of Fig 1 is identical to the AMM7 area, or if this is a subset?

Authors: It is a subset and has been added to the text (line 116-117).

Line 90: AMM7 is called a model, but perhaps "model setup" is more precise?

Authors: Done as suggested (line 95).

Line 93: Here the term "tracer" is used. It should be made clear whether tracer and particles are the same thing in this study.

Authors: Thanks for the hint. The use of "tracer" and "particle" should not be mixed up. Thus, Lagrangian particles have been defined (line 81-82) as well as the model tracers (T and S; line 102).

Line 95: Please provide a reference or justification for the choice of eddy diffusivity. It should be commented that this is constant throughout the area (which is not true in reality).

Authors: Please see, e.g. O'Dea et al. 2012 for a comparable setup of NEMO. The constant value of eddy diffusivity has been mentioned in line 104.

Line 96: Eddy viscosity should be a positive number.

Authors: Please keep in mind that we use biharmonic, not Laplacian mixing.

Section 2.2: More information should be given about the drifter type/characteristics/name, as near-surface drifters are affected by a varying degree of Stokes drift and wind drag, see e.g. Röhrs, J., K. H. Christensen, L. R. Hole, G. Broström, M. Drivdal, and S. Sundby (2012), Observation-based evaluation of surface wave effects on currents and trajectory forecasts, Ocean Dyn., 62, 1519–1533

Thus, a missing contribution from Stokes drift can possibly explain why the model currents are too slow in the comparison. Alternatively, SVP drifters (15m depth) from the Global Drifter Program could be used to validate the model current, so that Stokes drift would not be an

issue. Also a plot of the complete drifter trajectories should be shown, to justify whether they cover a substantial part of the area, or just locally to their deployment location.

Authors: In this paper the focus is on analysing Lagrangian trajectories (no real drifters). As far as Stokes drift is concerned, please see our earlier publication (Röhrs et al. (2012) and Stanev et al. (2019) are now cited in line 324-325) as well as the one discussing technical details about real drifters by Callies et al. (2017) (line 132). The restriction to the German Bight is given in line 136.

Callies, U., Groll, N., Horstmann, J., Kapitza, H., Klein, H., Maßmann, S., & Schwichtenberg, F. (2017). Surface drifters in the German Bight: model validation considering windage and Stokes drift. *Ocean Science*, *13*(5), 799–827. https://doi.org/10.5194/os-13-799-2017

Line 148/Table1: The number of comparison points should be provided.

Authors: Added in the Table. Note the changed order of the Tables.

Section 2.3. This discussion is a bit messy, and does also belong in the results section, rather than under "material and methods".

Authors: All results of Sect. 2.2 and 2.3 have been shifted to the results (note the rearranged order of the manuscript).

Line 184: It could be made clear (the first time) that Figure S2a refers to figure 2a in the supplements.

Authors: Done as suggested (line 252-253).

Line 207: could be commented that the Molinari and Kirway study is for the Caribbean during summer, thus quite different conditions.

Authors: This is correct and has been added (line 301).

Line 240: It should be commented (and discussed) that vertical mixing is not included.

Authors: The neglection of vertical mixing and how the movement of particles can be interpreted is added to the text.

Line 242: Should also be mentioned here that particles are released over the whole domain.

Authors: Done as suggested.

Line 244-246: The seeding locations of CR-V should also be shown on a figure

Authors: Showing the initial positions horizontally would require an own figure. Due to the amount of figures we decided to not add another one but to show these positions exemplarily for the lowest depth layer in Fig. 9e. We also improved the text accordingly (line 175-177).

Line 248: It should be mentioned explicitly that a separate offline trajectory model has to be used for the backwards simulations, as this is not possible to do with online simulations. However, a forward simulation with this offline model should also be done to benchmark it against the online forward simulations.

Authors: Such comparison has been made during the preparation of the manuscript and is mentioned in the text in line 151-153. The comparison in terms of NCPD using the results from

an online run without vertical advection and the same setup in OpenDrift for January 2015 is shown in Fig. R2.2 (NCPD online minus offline). The differences are rather minor. In text, the necessity of an offline model has been added (line 151).

[Figure]

**Fig. R2.2**. Surface January 2015 NCPD online 2-D (left), offline (middle) and the difference online minus offline (right).

Lines 274-279: What would be the difference between "density trend" and "residence time"?

Authors: Residence time (RT) is defined as $\bar{t}_w = V_S^0 / \bar{J}_V^0$ where $V_S^0$ is the total volume in the ocean reservoir and $\bar{J}_V^0$ is the mean flux through the reservoir in unit time in case of a steady state (superscripted 0); see, e.g. Whitfield (1979). It measures the time needed to completely replace the volume of water in a certain oceanic region. If the RT is referred to an individual water element Y, the RT formula can be rewritten as $\bar{t}_Y = \mathbf{Y}_S^0 / \bar{J}_Y^0$, where $\mathbf{Y}_S^0$ is the total mass of Y and $\bar{J}_Y^0$ is its flux through the reservoir. In our study, Y can be interpreted as a particle. Then, NCPD would be the ratio of $\bar{J}_Y^0$ and $\bar{J}_{Y,U=0}^0$ with (u,v,w) = 0, because they are the sum of particles over a certain period of time. Although both fluxes have the unit [mass/time], the latter flux could be interpreted as $\mathbf{Y}_S^0$, because it is constant in time. With this interpretation, NCPD is $1/\bar{t}_Y$ with the unit [1/month]. That is, NCPD is proportional to the inverse RT (line 227-228).

Whitfield, M. (1979). The mean oceanic residence time (MORT) concept - a rationalisation. *Marine Chemistry*, *8*(2), 101–123. https://doi.org/10.1016/0304-4203(79)90010-0

Line 278: "motionless situation" is a bit unclear, please rewrite sentence.

Authors: Done as suggested (line 215-217).

Line 302-304: Please clarify what is meant here.

Authors: Done (line 352-353).

Line 461: extra space after "Channel"

Authors: Removed.

Section 3 is a bit lengthy, and hard to read due to jumping back and forth between the experiments and referring to many figures. Making it a bit more compact and structured would help.

Authors: We tried to split the sections into equally long sections as well as in a Eulerian and a Lagrangian results part. We also reordered the experiments according to their appearance in the text; same for the supplementary figures. The jumping between experiments and figures results from a manuscript structure which is based on certain physical topics. Therefore, it is inevitable to refer only to one experiment. The figure references are thought to help to orientate while jumping back and forth. We still find them helpful and decided to keep them.

Figures

There are a lot of composite figures/maps of the area of interest. These are quite small and hard to read when printed on A4 paper. Could whitespace be reduced somehow?

Authors: We reduced the white space as much as possible, especially in Fig. 2. Insets have been enlarged. All labels should be readable now.

In the figure captions, the letters a), b)... should rather be placed before the explanation, and not after

Authors: Done as suggested.

Figure 2: CR and NTE should be written explicitly as "control run" and "no tides experiment", so that the figure can be read and understood also before reading the main text. Same for other figures.

Authors: Good idea. Done as suggested. Same for NCPD.

Line 847: und -> and

Authors: Changed.

Figure 3: a bit much spaghetti here, perhaps use even fewer than every 5th trajectory?

Authors: We changed it to every 8$^{th}$ particle. For us, the present figure is a good compromise between visualising the currents for both the surface and bottom as well as covering most of the domain with trajectories. We also remark, that we will provide this figure in high quality to OS, so that the reader can zoom in and see specific details.

Figure 4: Caption is quite hard to read. The '+' and '-' symbols are presumably placed "by hand"? This is generally ok, but they are quite many, and sometimes slightly displaced, perhaps to avoid overlap? So in practice I don't think these symbols work very well here. Could the point be visualized by another, more objective measure?

Authors: For Fig. 4, NCPD can be interpreted as a quantitative measure for particle accumulation. Thus, we decided to avoid any of these markers and we emphasise, that we describe examples of pronounced features (line 373-375 and 565).

Figure 5: Title of lower figure is "monthly average", but I guess it should be "yearly average", or "average of months"

Authors: This is correct and has been changed accordingly.

References

Please update this reference, where you refer to a discussion paper: Dagestad, K.- F., Röhrs, J., Breivik, Ø., and Ådlandsvik, B.: OpenDrift v1.0: a generic framework for trajectory modelling, Geosci. Model Dev., 11, 1405–1420, https://doi.org/10.5194/gmd-11-1405-2018, 2018.

Authors: Done as suggested.

---

## Author Response (AR2)

Dear Editor,

Below you find the reply to referee #1 followed by the marked-up manuscript. As before, our replies to the referee are in blue and lines numbers refer to related changes in the revised manuscript. Changes in the manuscript are in red.

Kind regards,
Marcel Ricker

**Reply to:**

**Report of revised manuscript "Circulation of the European Northwest Shelf: A Lagrangian perspective" by Marcel Ricker and Emil V. Stanev**

**Anonymous Referee #1**

Second review of 'Circulation of the European Northwest Shelf: A Lagrangian perspective' by Marcel Ricker and Emil Stanev.

We thank reviewer #1 for the second review of our paper. We provide point-by-point answers in the attached pdf.

I have read the revised document, and found it significantly improved. It can be accepted with minor changes.

I have two slightly more substantial points:

1. It is customary to present differences as 'scenario minus control'. The authors do the reverse, which makes it more difficult for me (and presumably many other readers) to interpret the plots/results. For instance, if tides are removed, and this results in a decrease in particle density somewhere, the difference plots show a positive value and vice versa. It is now clearly indicated how it was done, so not necessarily a show-stopper, but I would urge the authors to change this to improve the readability. It will likely also improve the narrative.

Authors: We follow the suggestion of the reviewer and show in the revised manuscript "scenario minus control" in the difference plots. This concerns Fig. 2h as well as Fig. 6, which have been updated. The text and captions have been changed accordingly.

2. The conclusions are very much formulated in terms of the fate of particles. However, the particles (in this study) are a means to an end, rather than the purpose of the study, which is to learn more about the ocean dynamics. Should the conclusions be not refrased? In the very least, I think the authors should add a few lines to the conclusions on what they have learned about their first objective (l. 55).

Authors: To answer this comment, we re-defined the third objective of the paper as the overall objective, while the first and second one were re-defined as the first and second specific objectives (lines 55, 65 and 70). Further, the outcome of the first specific objective is added in the Conclusions (line 653-654).

I also have a few more minor remarks/suggestions/typos/grammar, which I list below.

l. 18. persist in yearly

Authors: Done as suggested.

l. 22. vertical velocities in

Authors: Done as suggested.

l. 31. contributions from

Authors: Done as suggested.

l. 34. has not yet been widely investigated.

Authors: Done as suggested.

l. 42. around the southwest of Ireland

Authors: Done as suggested.

l. 60. the fronts. Freshwater

Authors: Done as suggested.

l. 84. of the velocity field

Authors: Done as suggested.

l. 100. by tides, and eddies

Authors: Done as suggested.

Authors: In the revised manuscript we refer to the publication of Stanev and Ricker (2020) for further details of model forcing and parameters. We also did the following changes and adapted the *Code/data availability* section accordingly (line 662-665):

l. 111. atmospheric model: please give the name and reference(s).

Authors: Added (line 111).

l. 113. climatological river runoff: please give the source of these data and reference(s).

Authors: Added (line 113-114).

l. 113. tidal forcing: please give the source of these data and reference(s).

Authors: Added (line 114-119).

l. 113. Please also provide information on the open boundary conditions for temperature and salinity.

Authors: Added (line 119-121).

l. 114, l. 128. 1 January

Authors: Changed throughout the manuscript.

l. 135. to reduce the effect of direct wind drag

Authors: Done as suggested.

l. 153. 2-D: horizontal?

Authors: Done as suggested.

l. 174. stripe: I understand that in German, both strip and stripe are translated as 'Streife'. In English, however, the two are subtly different. A stripe typically has (is!) a different colour from the surroundings. A strip can be a composed of a different material, or can be just a designation. So at this location, 'strip' should be used. There are other locations in the manuscript where 'stripe' can be used – please check carefully throughout.

Authors: Thanks for the very helpful explanation. We replaced "stripe" by "strip" throughout the manuscript where it not refers to a colour in a figure.

l. 184. particles were traced for

Authors: Done as suggested.

Figure 2b: change the text next to the colorbar into 'velocity amplitude'

Authors: Done as suggested.

Figure 2. It is still not clear from the document if the plotted quantities are for the surface, the bottom or depth-averaged. Please state this explicitly in the caption.

Authors: The caption already reads "Simulated surface properties …".

l. 258. Remove 'Dutch'. The Netherlands has no coasts in the German Bight.

Authors: Done as suggested.

l. 267. Atlantic water into the

Authors: Done as suggested.

l. 268. from satellite observations (Pietrzak et al., 2011) [they used the observations, but don't own them]

Authors: Done as suggested.

l. 269. East Anglian Plume

Authors: There are several spellings of this feature, e.g. East Anglian Plume, East-Anglia Plume, or East Anglian plume. We decided to keep our spelling, which is regularly used, e.g. by Pietrzak et al. (2011).

l. 279. Overall, Fig. 2a-f support

Authors: Done as suggested.

l. 356. 12.42 h: The figure says 12 h. Please change and make consistent.

Authors: Done as suggested.

l. 383. the continental slope.

Authors: Done as suggested.

l. 391. which drive particles away from the western

Authors: Done as suggested.

l. 394. occurs mainly and south: there's a bit of sentence missing here?

Authors: Word added (line 403).

l. 396. 'narrow channel': has a name: the Silver Pit

Authors: See next comment.

l. 397. 'basin to its southeast': has a name: the Oyster Grounds

Authors: "Silver Pit" and Oyster Ground" have been added in the text (line 404-405) and in Fig. 1.

l. 407. remove: therein

Authors: Done as suggested.

l. 410. The instantaneous particle positions (??)

Authors: Done as suggested.

Figure 5. Please replace 'monthly average' by 'Annual mean'.

Authors: Done as suggested.

Conclusions: as each of these could be cited/taken out of context by others, it should be stated specifically that they hold for neutrally buoyant particles (particles with different properties may well give different results). So:

l. 619: accumulate neutrally buoyant particles l. 625. accumulation areas for neutrally buoyant particles l. 630. remove neutrally buoyant particles l. 632. patterns for neutrally buoyant particles on the

Authors: We changed to "neutrally buoyant particles" at several places (e.g. line 16 and 624-625), which, in our opinion, makes clear about what we are talking.

To be completely accurate, the caveat that vertical turbulent diffusion of particles was neglected should be reiterated in the conclusions section.

**Authors**: This fact has been added (line 624-625).

l. 642. regimes at the 10 km scale. [The model does not resolve eddies slightly above and smaller than the horizontal resolution. These, however, do exist, especially near fronts and changes in topography. So this result may well be (or is very likely), at least to some extent, a model artifact!].

**Authors**: Added in line 652. What the model can resolve is mentioned in the part describing the model. We referred for more details to Stanev and Ricker (2020) where the issue about what the (same) model resolves and does not resolve is addressed in detail.

[revised manuscript text omitted]

**Fig. S4**. Scatter plots of (a, b) GPS drifter (May to July 2015) and control run (CR, see Table 1) surface velocities as well as (c, d) HF radar velocities of their (a, c) u and (b, d) v velocity components. Model and HF radar velocities were trilinearly (space and time) interpolated to drifter positions. The dashed line is the diagonal and denotes the optimal dot positions. The black dotted line is the quantile-quantile plot (qq-plot). The amount of available HF radar is less than the drifter data; thus these dots are enlarged. Statistics of each plot are shown in Table 2.

[Figure]

**Fig. S5**. Tendency of accumulation of surface released particles in the control run (CR, see Table 1) shown as NCPD for every month in 2015; (a) January to (l) December. Note the variability between the single months and compare with its mean in Fig. 5c.

[Figure]

**Fig. S6**. Tendency of accumulation of bottom released particles in the control run (CR, see Table 1) shown as NCPD for every month in 2015; (a) January to (l) December. Note the variability between the single months and compare with its mean in Fig. 5d.